# Stochastic Nested Variance Reduction for Nonconvex Optimization

**Dongruo Zhou**
Department of Computer Science
University of California, Los Angeles
Los Angeles, CA 90095
drzhou@cs.ucla.edu

**Pan Xu**
Department of Computer Science
University of California, Los Angeles
Los Angeles, CA 90095
panxu@cs.ucla.edu

**Quanquan Gu**
Department of Computer Science
University of California, Los Angeles
Los Angeles, CA 90095
qgu@cs.ucla.edu

## Abstract

We study finite-sum nonconvex optimization problems, where the objective function is an average of $n$ nonconvex functions. We propose a new stochastic gradient descent algorithm based on nested variance reduction. Compared with conventional stochastic variance reduced gradient (SVRG) algorithm that uses two reference points to construct a semi-stochastic gradient with diminishing variance in each iteration, our algorithm uses $K + 1$ nested reference points to build a semi-stochastic gradient to further reduce its variance in each iteration. For smooth nonconvex functions, the proposed algorithm converges to an $\epsilon$-approximate first-order stationary point (i.e., $\|\nabla F(\mathbf{x})\|_2 \leq \epsilon$) within $\widetilde{O}(n \wedge \epsilon^{-2} + \epsilon^{-3} \wedge n^{1/2}\epsilon^{-2})$[1] number of stochastic gradient evaluations. This improves the best known gradient complexity of SVRG $O(n + n^{2/3}\epsilon^{-2})$ and that of SCSG $O(n \wedge \epsilon^{-2} + \epsilon^{-10/3} \wedge n^{2/3}\epsilon^{-2})$. For gradient dominated functions, our algorithm also achieves better gradient complexity than the state-of-the-art algorithms. Thorough experimental results on different nonconvex optimization problems back up our theory.

## 1 Introduction

We study the following nonconvex finite-sum problem

$$\min_{\mathbf{x} \in \mathbb{R}^d} F(\mathbf{x}) := \frac{1}{n}\sum_{i=1}^{n} f_i(\mathbf{x}), \tag{1.1}$$

where each component function $f_i : \mathbb{R}^d \to \mathbb{R}$ has $L$-Lipschitz continuous gradient but may be nonconvex. A lot of machine learning problems fall into (1.1) such as empirical risk minimization (ERM) with nonconvex loss. Since finding the global minimum of (1.1) is general NP-hard [17], we instead aim at finding an $\epsilon$-approximate stationary point $\mathbf{x}$, which satisfies $\|\nabla F(\mathbf{x})\|_2 \leq \epsilon$, where $\nabla F(\mathbf{x})$ is the gradient of $F(\mathbf{x})$ at $\mathbf{x}$, and $\epsilon > 0$ is the accuracy parameter.

In this work, we mainly focus on first-order algorithms, which only need the function value and gradient evaluations. We use *gradient complexity*, the number of stochastic gradient evaluations,

to measure the convergence of different first-order algorithms.[2] For nonconvex optimization, it is well-known that *Gradient Descent* (GD) can converge to an $\epsilon$-approximate stationary point with $O(n \cdot \epsilon^{-2})$ [32] number of stochastic gradient evaluations. It can be seen that GD needs to calculate the full gradient at each iteration, which is a heavy load when $n \gg 1$. *Stochastic Gradient Descent* (SGD) has $O(\epsilon^{-4})$ gradient complexity to an $\epsilon$-approximate stationary point under the assumption that the stochastic gradient has a bounded variance [15]. While SGD only needs to calculate a minibatch of stochastic gradients in each iteration, due to the noise brought by stochastic gradients, its gradient complexity has a worse dependency on $\epsilon$. In order to improve the dependence of the gradient complexity of SGD on $n$ and $\epsilon$ for nonconvex optimization, variance reduction technique was firstly proposed in [41, 19, 46, 10, 30, 6, 45, 11, 16] for convex finite-sum optimization. Representative algorithms include Stochastic Average Gradient (SAG) [41], Stochastic Variance Reduced Gradient (SVRG) [19], SAGA [10], Stochastic Dual Coordinate Ascent (SDCA) [45], Finito [11] and Batching SVRG [16], to mention a few. The key idea behind variance reduction is that the gradient complexity can be saved if the algorithm use history information as *reference*. For instance, one representative variance reduction method is SVRG, which is based on a semi-stochastic gradient that is defined by two reference points. Since the the variance of this semi-stochastic gradient will diminish when the iterate gets closer to the minimizer, it therefore accelerates the convergence of stochastic gradient method. Later on, Harikandeh et al. [16] proposed Batching SVRG which also enjoys fast convergence property of SVRG without computing the full gradient. The convergence of SVRG under nonconvex setting was first analyzed in [13, 44], where $F$ is still convex but each component function $f_i$ can be nonconvex. The analysis for the general nonconvex function $F$ was done by [38, 5], which shows that SVRG can converge to an $\epsilon$-approximate stationary point with $O(n^{2/3} \cdot \epsilon^{-2})$ number of stochastic gradient evaluations. This result is strictly better than that of GD. Recently, Lei et al. [26] proposed a *Stochastically Controlled Stochastic Gradient* (SCSG) based on variance reduction, which further reduces the gradient complexity of SVRG to $O(n \wedge \epsilon^{-2} + \epsilon^{-10/3} \wedge (n^{2/3}\epsilon^{-2}))$. This result outperforms both GD and SGD strictly. To the best of our knowledge, this is the state-of-art gradient complexity under the smoothness (i.e., gradient lipschitz) and bounded stochastic gradient variance assumptions. A natural and long standing question is:

*Is there still room for improvement in nonconvex finite-sum optimization without making additional assumptions beyond smoothness and bounded stochastic gradient variance?*

In this paper, we provide an affirmative answer to the above question, by showing that the dependence on $n$ in the gradient complexity of SVRG [38, 5] and SCSG [26] can be further reduced. We propose a novel algorithm namely *Stochastic Nested Variance-Reduced Gradient descent* (SNVRG). Similar to SVRG and SCSG, our proposed algorithm works in a multi-epoch way. Nevertheless, the technique we developed is highly nontrivial. At the core of our algorithm is the multiple reference points-based variance reduction technique in each iteration. In detail, inspired by SVRG and SCSG, which uses two reference points to construct a semi-stochastic gradient with diminishing variance, our algorithm uses $K + 1$ reference points to construct a semi-stochastic gradient, whose variance decays faster than that of the semi-stochastic gradient used in SVRG and SCSG.

## 1.1 Our Contributions

Our major contributions are summarized as follows:

- We propose a stochastic nested variance reduction technique for stochastic gradient method, which reduces the dependence of the gradient complexity on $n$ compared with SVRG and SCSG.

- We show that our proposed algorithm is able to achieve an $\epsilon$-approximate stationary point with $\widetilde{O}(n \wedge \epsilon^{-2} + \epsilon^{-3} \wedge n^{1/2}\epsilon^{-2})$ stochastic gradient evaluations, which outperforms all existing first-order algorithms such as GD, SGD, SVRG and SCSG.

- As a by-product, when $F$ is a $\tau$-gradient dominated function, a variant of our algorithm can achieve an $\epsilon$-approximate global minimizer (i.e., $F(\mathbf{x}) - \min_{\mathbf{y}} F(\mathbf{y}) \leq \epsilon$) within $\widetilde{O}\big(n \wedge \tau\epsilon^{-1} + \tau(n \wedge \tau\epsilon^{-1})^{1/2}\big)$ stochastic gradient evaluations, which also outperforms the state-of-the-art.

## 1.2 Additional Related Work

Since it is hardly possible to review the huge body of literature on convex and nonconvex optimization due to space limit, here we review some additional most related work on accelerating nonconvex (finite-sum) optimization.

**Acceleration by high-order smoothness assumption** With only Lipschitz continuous gradient assumption, Carmon et al. [9] showed that the lower bound for both deterministic and stochastic algorithms to achieve an $\epsilon$-approximate stationary point is $\Omega(\epsilon^{-2})$. With high-order smoothness assumptions, i.e., Hessian Lipschitzness, Hessian smoothness etc., a series of work have shown the existence of acceleration. For instance, Agarwal et al. [1] gave an algorithm based on Fast-PCA which can achieve an $\epsilon$-approximate stationary point with gradient complexity $\widetilde{O}(n\epsilon^{-3/2} + n^{3/4}\epsilon^{-7/4})$ Carmon et al. [7, 8] showed two algorithms based on finding exact or inexact negative curvature which can achieve an $\epsilon$-approximate stationary point with gradient complexity $\widetilde{O}(n\epsilon^{-7/4})$. In this work, we

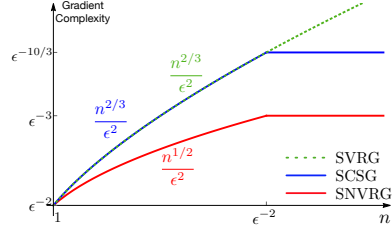

Figure 1: Comparison of gradient complexities.

only consider gradient Lipschitz without assuming Hessian Lipschitz or Hessian smooth. Therefore, our result is not directly comparable to the methods in this category.

**Acceleration by momentum** The fact that using momentum is able to accelerate algorithms has been shown both in theory and practice in convex optimization [35, 31, 18, 23, 14, 32, 29, 2]. However, there is no evidence that such acceleration exists in nonconvex optimization with only Lipschitz continuous gradient assumption [15, 27, 34, 28, 24]. If $F$ satisfies $\lambda$-strongly nonconvex, i.e., $\nabla^2 F \succeq -\lambda\mathbf{I}$, Allen-Zhu [3] proved that Natasha 1, an algorithm based on nonconvex momentum, is able to find an $\epsilon$-approximate stationary point in $\widetilde{O}(n^{2/3}L^{2/3}\lambda^{1/3}\epsilon^{-2})$. Later, Allen-Zhu [3] further showed that Natasha 2, an online version of Natasha 1, is able to achieve an $\epsilon$-approximate stationary point within $\widetilde{O}(\epsilon^{-3.25})$ stochastic gradient evaluations[3].

After our paper was submitted to NIPS and released on arXiv, a paper [12] was released on arXiv after our work, which independently proposes a different algorithm and achieves the same convergence rate for finding an $\epsilon$-approximate stationary point.

To give a thorough comparison of our proposed algorithm with existing first-order algorithms for nonconvex finite-sum optimization, we summarize the gradient complexity of the most relevant algorithms in Table 1. We also plot the gradient complexities of different algorithms in Figure 1 for nonconvex smooth functions. Note that GD and SGD are always worse than SVRG and SCSG according to Table 1. In addition, GNC-AGD and Natasha2 needs additional Hessian Lipschitz condition. Therefore, we only plot the gradient complexity of SVRG, SCSG and our proposed SNVRG in Figure 1.

**Notation:** Let $\mathbf{A} = [A_{ij}] \in \mathbb{R}^{d \times d}$ be a matrix and $\mathbf{x} = (x_1, ..., x_d)^\top \in \mathbb{R}^d$ be a vector. $\mathbf{I}$ denotes an identity matrix. We use $\|\mathbf{v}\|_2$ to denote the 2-norm of vector $\mathbf{v} \in \mathbb{R}^d$. We use $\langle \cdot, \cdot \rangle$ to represent the inner product of two vectors. Given two sequences $\{a_n\}$ and $\{b_n\}$, we write $a_n = O(b_n)$ if there exists a constant $0 < C < +\infty$ such that $a_n \leq C b_n$. We write $a_n = \Omega(b_n)$ if there exists a constant $0 < C < +\infty$, such that $a_n \geq C b_n$. We use notation $\widetilde{O}(\cdot)$ to hide logarithmic factors. We also make use of the notation $f_n \lesssim g_n$ ($f_n \gtrsim g_n$) if $f_n$ is less than (larger than) $g_n$ up to a constant. We use productive symbol $\prod_{i=a}^{b} c_i$ to denote $c_a c_{a+1} \ldots c_b$. Moreover, if $a > b$, we take the product as 1. We use $\lfloor \cdot \rfloor$ as the floor function. We use $\log(x)$ to represent the logarithm of $x$ to base 2. $a \wedge b$ is a shorthand notation for $\min(a, b)$.

## 2 Preliminaries

In this section, we present some definitions that will be used throughout our analysis.

Table 1: Comparisons on gradient complexity of different algorithms. The second column shows the gradient complexity for a nonconvex and smooth function to achieve an $\epsilon$-approximate stationary point (i.e., $\|\nabla F(\mathbf{x})\|_2 \leq \epsilon$). The third column presents the gradient complexity for a gradient dominant function to achieve an $\epsilon$-approximate global minimizer (i.e., $F(\mathbf{x}) - \min_\mathbf{x} F(\mathbf{x}) \leq \epsilon$). The last column presents the space complexity of all algorithms.

| Algorithm | nonconvex | gradient dominant | Hessian Lipschitz |
|---|---|---|---|
| GD | $O\left(\frac{n}{\epsilon^2}\right)$ | $\widetilde{O}(\tau n)$ | No |
| SGD | $O\left(\frac{1}{\epsilon^4}\right)$ | $O\left(\frac{1}{\epsilon^4}\right)$ | No |
| SVRG [38] | $O\left(\frac{n^{2/3}}{\epsilon^2}\right)$ | $\widetilde{O}(n + \tau n^{2/3})$ | No |
| SCSG [26] | $O\left(\frac{1}{\epsilon^{10/3}} \wedge \frac{n^{2/3}}{\epsilon^2}\right)$ | $\widetilde{O}\left(n \wedge \frac{\tau}{\epsilon} + \tau\left(n \wedge \frac{\tau}{\epsilon}\right)^{2/3}\right)$ | No |
| GNC-AGD [8] | $\widetilde{O}\left(\frac{n}{\epsilon^{1.75}}\right)$ | N/A | Needed |
| Natasha 2 [3] | $\widetilde{O}\left(\frac{1}{\epsilon^{3.25}}\right)$ | N/A | Needed |
| SNVRG (this paper) | $\widetilde{O}\left(\frac{1}{\epsilon^3} \wedge \frac{n^{1/2}}{\epsilon^2}\right)$ | $\widetilde{O}\left(n \wedge \frac{\tau}{\epsilon} + \tau\left(n \wedge \frac{\tau}{\epsilon}\right)^{1/2}\right)$ | No |

**Definition 2.1.** A function $f$ is $L$-smooth, if for any $\mathbf{x}, \mathbf{y} \in \mathbb{R}^d$, we have

$$\|\nabla f(\mathbf{x}) - \nabla f(\mathbf{y})\|_2 \leq L\|\mathbf{x} - \mathbf{y}\|_2. \tag{2.1}$$

Definition 2.1 implies that if $f$ is $L$-smooth, we have for any $\mathbf{x}, \mathbf{h} \in \mathbb{R}^d$

$$f(\mathbf{x} + \mathbf{h}) \leq f(\mathbf{x}) + \langle \nabla f(\mathbf{x}), \mathbf{h} \rangle + \frac{L}{2}\|\mathbf{h}\|_2^2. \tag{2.2}$$

**Definition 2.2.** A function $f$ is $\lambda$-strongly convex, if for any $\mathbf{x}, \mathbf{y} \in \mathbb{R}^d$, we have

$$f(\mathbf{x} + \mathbf{h}) \geq f(\mathbf{x}) + \langle \nabla f(\mathbf{x}), \mathbf{h} \rangle + \frac{\lambda}{2}\|\mathbf{h}\|_2^2. \tag{2.3}$$

**Definition 2.3.** A function $F$ with finite-sum structure in (1.1) is said to have stochastic gradients with bounded variance $\sigma^2$, if for any $\mathbf{x} \in \mathbb{R}^d$, we have

$$\mathbb{E}_i\|\nabla f_i(\mathbf{x}) - \nabla F(\mathbf{x})\|_2^2 \leq \sigma^2, \tag{2.4}$$

where $i$ a random index uniformly chosen from $[n]$ and $\mathbb{E}_i$ denotes the expectation over such $i$.

$\sigma^2$ is called the upper bound on the variance of stochastic gradients [26].

**Definition 2.4.** A function $F$ with finite-sum structure in (1.1) is said to have averaged $L$-Lipschitz gradient, if for any $\mathbf{x}, \mathbf{y} \in \mathbb{R}^d$, we have

$$\mathbb{E}_i\|\nabla f_i(\mathbf{x}) - \nabla f_i(\mathbf{y})\|_2^2 \leq L^2\|\mathbf{x} - \mathbf{y}\|_2^2, \tag{2.5}$$

where $i$ is a random index uniformly chosen from $[n]$ and $\mathbb{E}_i$ denotes the expectation over the choice.

**Definition 2.5.** We say a function $f$ is lower-bounded by $f^*$ if for any $\mathbf{x} \in \mathbb{R}^d$, $f(\mathbf{x}) \geq f^*$.

We also consider a class of functions namely gradient dominated functions [36], which is formally defined as follows:

**Definition 2.6.** We say function $f$ is $\tau$-gradient dominated if for any $\mathbf{x} \in \mathbb{R}^d$, we have

$$f(\mathbf{x}) - f(\mathbf{x}^*) \leq \tau \cdot \|\nabla f(\mathbf{x})\|_2^2, \tag{2.6}$$

where $\mathbf{x}^* \in \mathbb{R}^d$ is the global minimum of $f$.

Note that gradient dominated condition is also known as the Polyak-Lojasiewicz (P-L) condition [36], and is not necessarily convex. It is weaker than strong convexity as well as other popular conditions that appear in the optimization literature [20].

**Algorithm 1** One-epoch-SNVRG($\mathbf{x}_0, F, K, M, \{T_l\}, \{B_l\}, B$)

---

1: **Input:** initial point $\mathbf{x}_0$, function $F$, loop number $K$, step size parameter $M$, loop parameters $T_l, l \in [K]$, batch parameters $B_l, l \in [K]$, base batch size $B > 0$.
2: $\mathbf{x}_0^{(l)} \leftarrow \mathbf{x}_0, \mathbf{g}_0^{(l)} \leftarrow 0, 0 \le l \le K$
3: Uniformly generate index set $I \subset [n]$ without replacement, $|I| = B$
4: $\mathbf{g}_0^{(0)} \leftarrow 1/B \sum_{i \in I} \nabla f_i(\mathbf{x}_0)$
5: $\mathbf{v}_0 \leftarrow \sum_{l=0}^K \mathbf{g}_0^{(l)}$
6: $\mathbf{x}_1 = \mathbf{x}_0 - 1/(10M) \cdot \mathbf{v}_0$
7: **for** $t = 1, ..., \prod_{l=1}^K T_l - 1$ **do**
8: $\quad r = \min\{j : 0 = (t \mod \prod_{l=j+1}^K T_l), 0 \le j \le K\}$
9: $\quad \{\mathbf{x}_t^{(l)}\} \leftarrow$ Update_reference_points($\{\mathbf{x}_{t-1}^{(l)}\}, \mathbf{x}_t, r$), $0 \le l \le K$.
10: $\quad \{\mathbf{g}_t^{(l)}\} \leftarrow$ Update_reference_gradients($\{\mathbf{g}_{t-1}^{(l)}\}, \{\mathbf{x}_t^{(l)}\}, r$), $0 \le l \le K$.
11: $\quad \mathbf{v}_t \leftarrow \sum_{l=0}^K \mathbf{g}_t^{(l)}$
12: $\quad \mathbf{x}_{t+1} \leftarrow \mathbf{x}_t - 1/(10M) \cdot \mathbf{v}_t$
13: **end for**
14: $\mathbf{x}_{\text{out}} \leftarrow$ uniformly random choice from $\{\mathbf{x}_t\}$, where $0 \le t < \prod_{l=1}^K T_l$
15: $T = \prod_{l=1}^K T_l$
16: **Output:** $[\mathbf{x}_{\text{out}}, \mathbf{x}_T]$

---

17: **Function:** Update_reference_points($\{\mathbf{x}_{\text{old}}^{(l)}\}, \mathbf{x}, r$)
18: $\mathbf{x}_{\text{new}}^{(l)} \leftarrow \mathbf{x}_{\text{old}}^{(l)}, 0 \le l \le r - 1; \mathbf{x}_{\text{new}}^{(l)} \leftarrow \mathbf{x}, r \le l \le K$
19: **return** $\{\mathbf{x}_{\text{new}}^{(l)}\}$

---

20: **Function:** Update_reference_gradients($\{\mathbf{g}_{\text{old}}^{(l)}\}, \{\mathbf{x}_{\text{new}}^{(l)}\}, r$)
21: $\mathbf{g}_{\text{new}}^{(l)} \leftarrow \mathbf{g}_{\text{old}}^{(l)}, 0 \le l < r$
22: **for** $r \le l \le K$ **do**
23: $\quad$ Uniformly generate index set $I \subset [n]$ without replacement, $|I| = B_l$
24: $\quad \mathbf{g}_{\text{new}}^{(l)} \leftarrow 1/B_l \sum_{i \in I} \left[ \nabla f_i(\mathbf{x}_{\text{new}}^{(l)}) - \nabla f_i(\mathbf{x}_{\text{new}}^{(l-1)}) \right]$
25: **end for**
26: **return** $\{\mathbf{g}_{\text{new}}^{(l)}\}$.

---

## 3 The Proposed Algorithm

In this section, we present our nested stochastic variance reduction algorithm, namely, SNVRG.

**One-epoch-SNVRG:** We first present the key component of our main algorithm, One-epoch-SNVRG, which is displayed in Algorithm 1. The most innovative part of Algorithm 1 attributes to the $K + 1$ *reference points* and $K + 1$ *reference gradients*. Note that when $K = 1$, Algorithm 1 reduces to one epoch of SVRG algorithm [19, 38, 5]. To better understand our One-epoch SNVRG algorithm, it would be helpful to revisit the original SVRG which is a special case of our algorithm. For the finite-sum optimization problem in (1.1), the original SVRG takes the following updating formula

$$\mathbf{x}_{t+1} = \mathbf{x}_t - \eta \mathbf{v}_t = \mathbf{x}_t - \eta \big( \nabla F(\widetilde{\mathbf{x}}) + \nabla f_{i_t}(\mathbf{x}_t) - \nabla f_{i_t}(\widetilde{\mathbf{x}}) \big),$$

where $\eta > 0$ is the step size, $i_t$ is a random index uniformly chosen from $[n]$ and $\widetilde{\mathbf{x}}$ is a snapshot for $\mathbf{x}_t$ after every $T_1$ iterations. There are two reference points in the update formula at $\mathbf{x}_t$: $\mathbf{x}_t^{(0)} = \widetilde{\mathbf{x}}$ and $\mathbf{x}_t^{(1)} = \mathbf{x}_t$. Note that $\widetilde{\mathbf{x}}$ is updated every $T_1$ iterations, namely, $\widetilde{\mathbf{x}}$ is set to be $\mathbf{x}_t$ only when $(t \mod T_1) = 0$. Moreover, in the semi-stochastic gradient $\mathbf{v}_t$, there are also two reference gradients and we denote them by $\mathbf{g}_t^{(0)} = \nabla F(\widetilde{\mathbf{x}})$ and $\mathbf{g}_t^{(1)} = \nabla f_{i_t}(\mathbf{x}_t) - \nabla f_{i_t}(\widetilde{\mathbf{x}}) = \nabla f_{i_t}(\mathbf{x}_t^{(1)}) - \nabla f_{i_t}(\mathbf{x}_t^{(0)})$.

Back to our One-epoch-SNVRG, we can define similar reference points and reference gradients as that in the special case of SVRG. Specifically, for $t = 0, \ldots, \prod_{l=1}^K T_l - 1$, each point $\mathbf{x}_t$ has $K + 1$

reference points $\{\mathbf{x}_t^{(l)}\}, l = 0, \ldots, K$, which is set to be $\mathbf{x}_t^{(l)} = \mathbf{x}_{t^l}$ with index $t^l$ defined as

$$t^l = \left\lfloor \frac{t}{\prod_{k=l+1}^{K} T_k} \right\rfloor \cdot \prod_{k=l+1}^{K} T_k. \tag{3.1}$$

Specially, note that we have $\mathbf{x}_t^{(0)} = \mathbf{x}_0$ and $\mathbf{x}_t^{(K)} = \mathbf{x}_t$ for all $t = 0, \ldots, \prod_{l=1}^{K} T_l - 1$. Similarly, $\mathbf{x}_t$ also has $K + 1$ reference gradients $\{\mathbf{g}_t^{(l)}\}$, which can be defined based on the reference points $\{\mathbf{x}_t^{(l)}\}$:

$$\mathbf{g}_t^{(0)} = \frac{1}{B} \sum_{i \in I} \nabla f_i(\mathbf{x}_0), \qquad \mathbf{g}_t^{(l)} = \frac{1}{B_l} \sum_{i \in I_l} \left[ \nabla f_i(\mathbf{x}_t^{(l)}) - \nabla f_i(\mathbf{x}_t^{(l-1)}) \right], l = 1, \ldots, K, \tag{3.2}$$

where $I, I_l$ are random index sets with $|I| = B, |I_l| = B_l$ and are uniformly generated from $[n]$ without replacement. Based on the reference points and reference gradients, we then update $\mathbf{x}_{t+1} = \mathbf{x}_t - 1/(10M) \cdot \mathbf{v}_t$, where $\mathbf{v}_t = \sum_{l=0}^{K} \mathbf{g}_t^{(l)}$ and $M$ is the step size parameter. The illustration of reference points and gradients of SNVRG is displayed in Figure 2.

We remark that it would be a huge waste for us to re-evaluate $\mathbf{g}_t^{(l)}$ at each iteration. Fortunately, due to the fact that each reference point is only updated after a long period, we can maintain $\mathbf{g}_t^{(l)} = \mathbf{g}_{t-1}^{(l)}$ and only need to update $\mathbf{g}_t^{(l)}$ when $\mathbf{x}_t^{(l)}$ has been updated as is suggested by Line 24 in Algorithm 1.

**SNVRG:** Using One-epoch-SNVRG (Algorithm 1) as a building block, we now present our main algorithm: Algorithm 2 for nonconvex finite-sum optimization to find an $\epsilon$-approximate stationary point. At each iteration of Algorithm 2, it executes One-epoch-SNVRG (Algorithm 1) which takes $\mathbf{z}_{s-1}$ as its input and outputs $[\mathbf{y}_s, \mathbf{z}_s]$. We choose $\mathbf{y}_{\text{out}}$ as the output of Algorithm 2 uniformly from $\{\mathbf{y}_s\}$, for $s = 1, \ldots, S$.

**SNVRG-PL:** In addition, when function $F$ in (1.1) is gradient dominated as defined in Definition 2.6 (P-L condition), it has been proved that the global minimum can be found by SGD [20], SVRG [38] and SCSG [26] very efficiently. Following a similar trick used in [38], we present Algorithm 3 on top of Algorithm 2, to find the global minimum in this setting. We call Algorithm 3 SNVRG-PL, because gradient dominated condition is also known as Polyak-Lojasiewicz (PL) condition [36].

**Space complexity**: We briefly compare the space complexity between our algorithms and other variance reduction based algorithms. SVRG and SCSG needs $O(d)$ space complexity to store one reference gradient, SAGA [10] needs to store reference gradients for each component functions, and its space complexity is $O(nd)$ without using any trick. For our algorithm SNVRG, we need to store $K$ reference gradients, thus its space complexity is $O(Kd)$. In our theory, we will show that $K = O(\log \log n)$. Therefore, the space complexity of our algorithm is actually $\widetilde{O}(d)$, which is almost comparable to that of SVRG and SCSG.

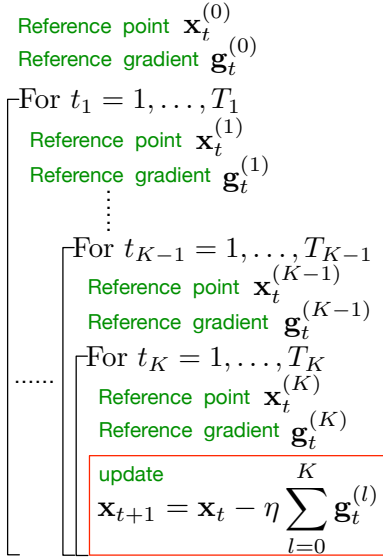

Figure 2: Illustration of reference points and gradients.

---

| **Algorithm 2** SNVRG | **Algorithm 3** SNVRG-PL |
|---|---|
| 1: **Input:** initial point $\mathbf{z}_0$, function $F$, $K$, $M$, $\{T_l\}$, $\{B_l\}$, batch $B$, $S$. | 1: **Input:** initial point $\mathbf{z}_0$, function $F$, $K$, $M$, $\{T_l\}$, $\{B_l\}$, batch $B$, $S$, $U$. |
| 2: **for** $s = 1, \ldots, S$ **do** | 2: **for** $u = 1, \ldots, U$ **do** |
| 3:    denote $\mathcal{P} = (F, K, M, \{T_l\}, \{B_l\}, B)$ | 3:    denote $\mathcal{Q} = (F, K, M, \{T_l\}, \{B_l\}, B, S)$ |
| 4:    $[\mathbf{y}_s, \mathbf{z}_s] \leftarrow$ One-epoch-SNVRG($\mathbf{z}_{s-1}, \mathcal{P}$) | 4:    $\mathbf{z}_u = $ SNVRG($\mathbf{z}_{u-1}, \mathcal{Q}$) |
| 5: **end for** | 5: **end for** |
| 6: **Output:** Uniformly choose $\mathbf{y}_{\text{out}}$ from $\{\mathbf{y}_s\}$. | 6: **Output:** $\mathbf{z}_{\text{out}} = \mathbf{z}_U$. |

---

# 4 Main Theory

In this section, we provide the convergence analysis of SNVRG.

## 4.1 Convergence of SNVRG

We first analyze One-epoch-SNVRG (Algorithm 1) and provide a particular choice of parameters.

**Lemma 4.1.** Suppose that $F$ has averaged $L$-Lipschitz gradient, in Algorithm 1, suppose $B \geq 2$ and let the number of nested loops be $K = \log \log B$. Choose the step size parameter as $M = 6L$. For the loop and batch parameters, let $T_1 = 2, B_1 = 6^K \cdot B$ and

$$T_l = 2^{2^{l-2}}, \qquad B_l = 6^{K-l+1} \cdot B/2^{2^{l-1}},$$

for all $2 \leq l \leq K$. Then the output of Algorithm 1 $[\mathbf{x}_{\text{out}}, \mathbf{x}_T]$ satisfies

$$\mathbb{E}\|\nabla F(\mathbf{x}_{\text{out}})\|_2^2 \leq C\left(\frac{L}{B^{1/2}} \cdot \mathbb{E}\big[F(\mathbf{x}_0) - F(\mathbf{x}_T)\big] + \frac{\sigma^2}{B} \cdot \mathbb{1}(B < n)\right) \tag{4.1}$$

within $1 \vee (7B \log^3 B)$ stochastic gradient computations, where $T = \prod_{l=1}^{K} T_l$, $C = 600$ is a constant and $\mathbb{1}(\cdot)$ is the indicator function.

The following theorem shows the gradient complexity for Algorithm 2 to find an $\epsilon$-approximate stationary point with a constant base batch size $B$.

**Theorem 4.2.** Suppose that $F$ has averaged $L$-Lipschitz gradient and stochastic gradients with bounded variance $\sigma^2$. In Algorithm 2, let $B = n \wedge (2C\sigma^2/\epsilon^2)$, $S = 1 \vee (2CL\Delta_F/(B^{1/2}\epsilon^2))$ and $C = 600$. The rest parameters $(K, M, \{B_l\}, \{T_l\})$ are chosen the same as in Lemma 4.1. Then the output $\mathbf{y}_{\text{out}}$ of Algorithm 2 satisfies $\mathbb{E}\|\nabla F(\mathbf{y}_{\text{out}})\|_2^2 \leq \epsilon^2$ with less than

$$O\left(\log^3\left(\frac{\sigma^2}{\epsilon^2} \wedge n\right)\left[\frac{\sigma^2}{\epsilon^2} \wedge n + \frac{L\Delta_F}{\epsilon^2}\left[\frac{\sigma^2}{\epsilon^2} \wedge n\right]^{1/2}\right]\right) \tag{4.2}$$

stochastic gradient computations, where $\Delta_F = F(\mathbf{z}_0) - F^*$.

**Remark 4.3.** If we treat $\sigma^2$, $L$ and $\Delta_F$ as constants, and assume $\epsilon \ll 1$, then (4.2) can be simplified to $\widetilde{O}(\epsilon^{-3} \wedge n^{1/2}\epsilon^{-2})$. This gradient complexity is strictly better than $O(\epsilon^{-10/3} \wedge n^{2/3}\epsilon^{-2})$, which is achieved by SCSG [26]. Specifically, when $n \lesssim 1/\epsilon^2$, our proposed SNVRG is faster than SCSG by a factor of $n^{1/6}$; when $n \gtrsim 1/\epsilon^2$, SNVRG is faster than SCSG by a factor of $\epsilon^{-1/3}$. Moreover, SNVRG also outperforms Natasha 2 [3] which attains $\widetilde{O}(\epsilon^{-3.25})$ gradient complexity and needs the additional Hessian Lipschitz condition.

## 4.2 Convergence of SNVRG-PL

We now consider the case when $F$ is a $\tau$-gradient dominated function. In general, we are able to find an $\epsilon$-approximate global minimizer of $F$ instead of only an $\epsilon$-approximate stationary point. Algorithm 3 uses Algorithm 2 as a component.

**Theorem 4.4.** Suppose that $F$ has averaged $L$-Lipschitz gradient and stochastic gradients with bounded variance $\sigma^2$, $F$ is a $\tau$-gradient dominated function. In Algorithm 3, let the base batch size $B = n \wedge (4C_1\tau\sigma^2/\epsilon)$, the number of epochs for SNVRG $S = 1 \vee (2C_1\tau L/B^{1/2})$ and the number of epochs $U = \log(2\Delta_F/\epsilon)$. The rest parameters $(K, M, \{B_l\}, \{T_l\})$ are chosen as the same in Lemma 4.1. Then the output $\mathbf{z}_{\text{out}}$ of Algorithm 3 satisfies $\mathbb{E}\big[F(\mathbf{z}_{\text{out}}) - F^*\big] \leq \epsilon$ within

$$O\left(\log^3\left(n \wedge \frac{\tau\sigma^2}{\epsilon}\right)\log\frac{\Delta_F}{\epsilon}\left[n \wedge \frac{\tau\sigma^2}{\epsilon} + \tau L\left[n \wedge \frac{\tau\sigma^2}{\epsilon}\right]^{1/2}\right]\right) \tag{4.3}$$

stochastic gradient computations, where $\Delta_F = F(\mathbf{z}_0) - F^*$

**Remark 4.5.** If we treat $\sigma^2$, $L$ and $\Delta_F$ as constants, then the gradient complexity in (4.3) turns into $\widetilde{O}(n \wedge \tau\epsilon^{-1} + \tau(n \wedge \tau\epsilon^{-1})^{1/2})$. Compared with nonconvex SVRG [39] which achieves $\widetilde{O}(n + \tau n^{2/3})$ gradient complexity, our SNVRG-PL is strictly better than SVRG in terms of the first summand and is faster than SVRG at least by a factor of $n^{1/6}$ in terms of the second summand. Compared with a more general variant of SVRG, namely, the SCSG algorithm [26], which attains $\widetilde{O}(n \wedge \tau\epsilon^{-1} + \tau(n \wedge \tau\epsilon^{-1})^{2/3})$ gradient complexity, SNVRG-PL also outperforms it by a factor of $(n \wedge \tau\epsilon^{-1})^{1/6}$.

If we further assume that $F$ is $\lambda$-strongly convex, then it is easy to verify that $F$ is also $1/(2\lambda)$-gradient dominated. As a direct consequence, we have the following corollary:

**Corollary 4.6.** Under the same conditions and parameter choices as Theorem 4.4. If we additionally assume that $F$ is $\lambda$-strongly convex, then Algorithm 3 will outputs an $\epsilon$-approximate global minimizer within

$$\widetilde{O}\left(n \wedge \frac{\lambda\sigma^2}{\epsilon} + \kappa \cdot \left[n \wedge \frac{\lambda\sigma^2}{\epsilon}\right]^{1/2}\right) \tag{4.4}$$

stochastic gradient computations, where $\kappa = L/\lambda$ is the condition number of $F$.

**Remark 4.7.** Corollary 4.6 suggests that when we regard $\lambda$ and $\sigma^2$ as constants and set $\epsilon \ll 1$, Algorithm 3 is able to find an $\epsilon$-approximate global minimizer within $\widetilde{O}(n + n^{1/2}\kappa)$ stochastic gradient computations, which matches SVRG-lep in Katyusha X [4]. Using catalyst techniques [29] or Katyusha momentum [2], it can be further accelerated to $\widetilde{O}(n + n^{3/4}\sqrt{\kappa})$, which matches the best-known convergence rate [43, 4].

## 5  Experiments

In this section, we compare our algorithm SNVRG with other baseline algorithms on training a convolutional neural network for image classification. We compare the performance of the following algorithms: *SGD*; SGD with momentum [37] (denoted by *SGD-momentum*); *ADAM*[21]; *SCSG* [26]. It is worth noting that *SCSG* is a special case of SNVRG when the number of nested loops $K = 1$. Due to the memory cost, we did not compare *GD* and *SVRG* which need to calculate the full gradient. Although our theoretical analysis holds for general $K$ nested loops, it suffices to choose $K = 2$ in *SNVRG* to illustrate the effectiveness of the nested structure for the simplification of implementation. In this case, we have 3 reference points and gradients. All experiments are conducted on Amazon AWS p2.xlarge servers which comes with Intel Xeon E5 CPU and NVIDIA Tesla K80 GPU (12G GPU RAM). All algorithm are implemented in Pytorch platform version 0.4.0 within Python 3.6.4.

**Datasets** We use three image datasets: (1) The MNIST dataset [42] consists of handwritten digits and has $50,000$ training examples and $10,000$ test examples. The digits have been size-normalized to fit the network, and each image is 28 pixels by 28 pixels. (2) CIFAR10 dataset [22] consists of images in 10 classes and has $50,000$ training examples and $10,000$ test examples. The digits have been size-normalized to fit the network, and each image is 32 pixels by 32 pixels. (3) SVHN dataset [33] consists of images of digits and has $531,131$ training examples and $26,032$ test examples. The digits have been size-normalized to fit the network, and each image is 32 pixels by 32 pixels.

**CNN Architecture** We use the standard LeNet [25], which has two convolutional layers with 6 and 16 filters of size 5 respectively, followed by three fully-connected layers with output size 120, 84 and 10. We apply max pooling after each convolutional layer.

**Implementation Details & Parameter Tuning** We did not use the random data augmentation which is set as default by Pytorch, because it will apply random transformation (e.g., clip and rotation) at the beginning of each epoch on the original image dataset, which will ruin the finite-sum structure of the loss function. We set our grid search rules for all three datasets as follows. For *SGD*, we search the batch size from $\{256, 512, 1024, 2048\}$ and the initial step sizes from $\{1, 0.1, 0.01\}$. For *SGD-momentum*, we set the momentum parameter as $0.9$. We search its batch size from $\{256, 512, 1024, 2048\}$ and the initial learning rate from $\{1, 0.1, 0.01\}$. For *ADAM*, we search the batch size from $\{256, 512, 1024, 2048\}$ and the initial learning rate from $\{0.01, 0.001, 0.0001\}$. For *SCSG* and *SNVRG*, we choose loop parameters $\{T_l\}$ which satisfy $B_l \cdot \prod_{j=1}^{l} T_j = B$ automatically. In addition, for *SCSG*, we set the batch sizes $(B, B_1) = (B, B/b)$, where $b$ is the batch size ratio parameter. We search $B$ from $\{256, 512, 1024, 2048\}$ and we search $b$ from $\{2, 4, 8\}$. We search its initial learning rate from $\{1, 0.1, 0.01\}$. For our proposed *SNVRG*, we set the batch sizes $(B, B_1, B_2) = (B, B/b, B/b^2)$, where $b$ is the batch size ratio parameter. We search $B$ from $\{256, 512, 1024, 2048\}$ and $b$ from $\{2, 4, 8\}$. We search its initial learning rate from $\{1, 0.1, 0.01\}$. Following the convention of deep learning practice, we apply learning rate decay schedule to each algorithm with the learning rate decayed by $0.1$ every 20 epochs. We also conducted experiments based on plain implementation of different algorithms without learning rate decay, which is deferred to the appendix.

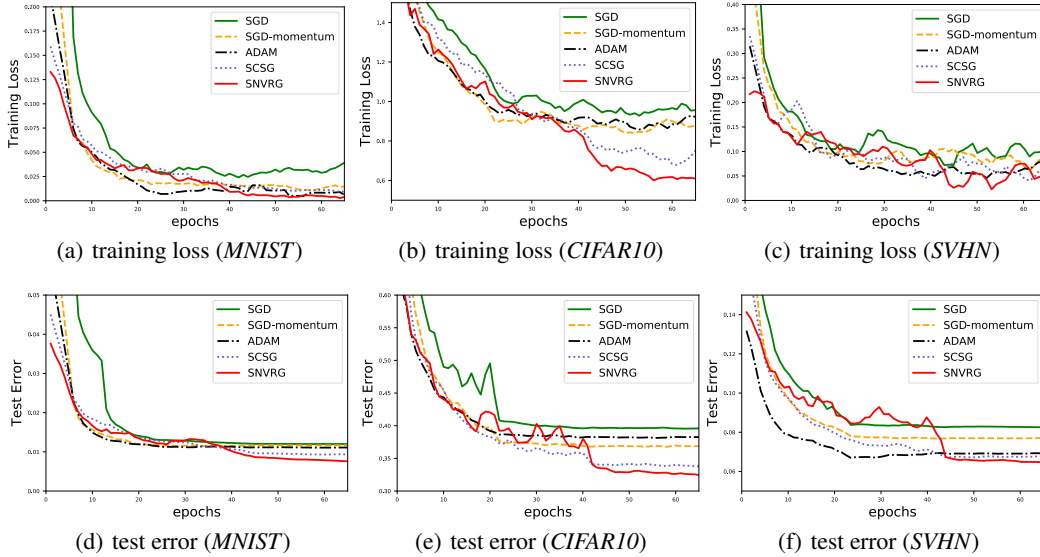

| | | |
|---|---|---|
| (a) training loss (*MNIST*) | (b) training loss (*CIFAR10*) | (c) training loss (*SVHN*) |
| (d) test error (*MNIST*) | (e) test error (*CIFAR10*) | (f) test error (*SVHN*) |

Figure 3: Experiment results on different datasets with learning rate decay. (a) and (d) depict the training loss and test error (top-1 error) v.s. data epochs for training LeNet on MNIST dataset. (b) and (e) depict the training loss and test error v.s. data epochs for training LeNet on CIFAR10 dataset. (c) and (f) depict the training loss and test error v.s. data epochs for training LeNet on SVHN dataset.

We plotted the training loss and test error for different algorithms on each dataset in Figure 3. The results on MNIST are presented in Figures 3(a) and 3(d); the results on CIFAR10 are in Figures 3(b) and 3(e); and the results on SVHN dataset are shown in Figures 3(c) and 3(f). It can be seen that with learning rate decay schedule, our algorithm *SNVRG* outperforms all baseline algorithms, which confirms that the use of nested reference points and gradients can accelerate the nonconvex finite-sum optimization.

We would like to emphasize that, while this experiment is on training convolutional neural networks, the major goal of this experiment is to illustrate the advantage of our algorithm and corroborate our theory, rather than claiming a state-of-the-art algorithm for training deep neural networks.

## 6 Conclusions and Future Work

In this paper, we proposed a stochastic nested variance reduced gradient method for finite-sum nonconvex optimization. It achieves substantially better gradient complexity than existing first-order algorithms. This partially resolves a long standing question that whether the dependence of gradient complexity on $n$ for nonconvex SVRG and SCSG can be further improved. There is still an open question: whether $\widetilde{O}(n \wedge \epsilon^{-2} + \epsilon^{-3} \wedge n^{1/2}\epsilon^{-2})$ is the optimal gradient complexity for finite-sum and stochastic nonconvex optimization problem? For finite-sum nonconvex optimization problem, the lower bound has been proved in Fang et al. [12], which suggests that our algorithm is near optimal up to a logarithmic factor. However, for general stochastic problem, the lower bound is still unknown. We plan to derive such lower bound in our future work. On the other hand, our algorithm can also be extended to deal with nonconvex nonsmooth finite-sum optimization using proximal gradient [40].

## Acknowledgement

We would like to thank the anonymous reviewers for their helpful comments. This research was sponsored in part by the National Science Foundation IIS-1652539 and BIGDATA IIS-1855099. We also thank AWS for providing cloud computing credits associated with the NSF BIGDATA award. The views and conclusions contained in this paper are those of the authors and should not be interpreted as representing any funding agencies.

## Footnotes

[1] $\widetilde{O}(\cdot)$ hides the logarithmic factors, and $a \wedge b$ means $\min(a, b)$.

[2]While we use gradient complexity as in [26] to present our result, it is basically the same if we use incremental first-order oracle (IFO) complexity used by [38]. In other words, these are directly comparable.

[3]In fact, Natasha 2 is guaranteed to converge to an $(\epsilon, \sqrt{\epsilon})$-approximate second-order stationary point with $\widetilde{O}(\epsilon^{-3.25})$ gradient complexity, which implies the convergence to an $\epsilon$-approximate stationary point.

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
