[Supplementary Material]

# A Proof of the Main Theoretical Results

In this section, we provide the proofs of our main theories in Section 4.

## A.1 Proof of Lemma 4.1

We first prove our key lemma on One-epoch-SNVRG. In order to prove Lemma 4.1, we need the following supporting lemma:

**Lemma A.1.** Let $T = \prod_{l=1}^{K} T_l$. If the step size and batch size parameters in Algorithm 1 satisfy $M \geq 6L$ and $B_l \geq 6^{K-l+1}(\prod_{s=l}^{K} T_s)^2$, then the output of Algorithm 1 satisfies

$$\mathbb{E}\|\nabla F(\mathbf{x}_{\text{out}})\|_2^2 \leq C\left(\frac{M}{T} \cdot \mathbb{E}\left[F(\mathbf{x}_0) - F(\mathbf{x}_T)\right] + \frac{2\sigma^2}{B} \cdot \mathbb{1}(B < n)\right), \tag{A.1}$$

where $C = 100$ is a constant.

*Proof of Lemma 4.1.* Note that $B = 2^{2^K}$, we can easily check that the choice of $M, \{T_l\}, \{B_l\}$ in Lemma 4.1 satisfies the assumption of Lemma A.1. Moreover, we have

$$T = \prod_{l=1}^{K} T_l = B^{1/2}. \tag{A.2}$$

We now submit (A.2) into (A.1), which immediately implies (4.1).

Next we compute how many stochastic gradient computations we need in total after we run One-epoch-SNVRG once. According to the update of reference gradients in Algorithm 1, we only update $\mathbf{g}_t^{(0)}$ once at the beginning of Algorithm 1 (Line 4), which needs $B$ stochastic gradient computations. For $\mathbf{g}_t^{(l)}$, we only need to update it when $0 = (t \mod \prod_{j=l+1}^{K} T_j)$, and thus we need to sample $\mathbf{g}_t^{(l)}$ for $T/\prod_{j=l+1}^{K} T_j = \prod_{j=1}^{l} T_j$ times. We need $2B_l$ stochastic gradient computations for each sampling procedure (Line 24 in Algorithm 1). We use $\mathcal{T}$ to represent the total number of stochastic gradient computations, then based on above arguments we have

$$\mathcal{T} = B + 2\sum_{l=1}^{K} B_l \cdot \prod_{j=1}^{l} T_j. \tag{A.3}$$

Now we calculate $\mathcal{T}$ under the parameter choice of Lemma 4.1. Note that we can easily verify the following results:

$$\prod_{j=1}^{l} T_j = 2^{2^{l-1}} = B^{\frac{2^l}{2^{K+1}}}, \quad B_1 \cdot \prod_{j=1}^{1} T_j = 2 \times 6^K B, \quad B_l \cdot \prod_{j=1}^{l} T_j = 6^{K-l+1} B. \tag{A.4}$$

Submit (A.4) into (A.3) yields the following results:

$$\begin{aligned}
\mathcal{T} &= B + 2\left(2 \times 6^K B + \sum_{l=2}^{K} 6^{K-l+1} B\right) \\
&< B + 6 \times 6^K B \\
&= B + 6 \times 6^{\log\log B} B \\
&< B + 6B \log^3 B. 
\end{aligned} \tag{A.5}$$

Therefore, the total gradient complexity $\mathcal{T}$ is bounded as follows.

$$\mathcal{T} \leq B + 6B \log^3 B \leq 7B \log^3 B. \tag{A.6}$$

$\square$

## A.2 Proof of Theorem 4.2

Now we prove our main theorem which spells out the gradient complexity of SNVRG.

*Proof of Theorem 4.2.* By (4.1) we have

$$\mathbb{E}\|\nabla F(\mathbf{y}_s)\|_2^2 \le C\left(\frac{L}{B^{1/2}} \cdot \mathbb{E}\big[F(\mathbf{z}_{s-1}) - F(\mathbf{z}_s)\big] + \frac{\sigma^2}{B} \cdot \mathbb{1}(B < n)\right), \qquad (A.7)$$

where $C = 600$. Taking summation for (A.7) over $s$ from 1 to $S$, we have

$$\sum_{s=1}^{S} \mathbb{E}\|\nabla F(\mathbf{y}_s)\|_2^2 \le C\left(\frac{L}{B^{1/2}} \cdot \mathbb{E}\big[F(\mathbf{z}_0) - F(\mathbf{z}_S)\big] + \frac{\sigma^2}{B} \cdot \mathbb{1}(B < n) \cdot S\right). \qquad (A.8)$$

Dividing both sides of (A.8) by $S$, we immediately obtain

$$\mathbb{E}\|\nabla F(\mathbf{y}_{\text{out}})\|_2^2 \le C\left(\frac{L\mathbb{E}\big[F(\mathbf{z}_0) - F^*\big]}{SB^{1/2}} + \frac{\sigma^2}{B} \cdot \mathbb{1}(B < n)\right) \qquad (A.9)$$

$$= C\left(\frac{L\Delta_F}{SB^{1/2}} + \frac{\sigma^2}{B} \cdot \mathbb{1}(B < n)\right), \qquad (A.10)$$

where (A.9) holds because $F(\mathbf{z}_S) \ge F^*$ and by the definition $\Delta_F = F(\mathbf{z}_0) - F^*$. By the choice of parameters in Theorem 4.2, we have $B = n \wedge (2C\sigma^2/\epsilon^2)$, $S = 1 \vee (2CL\Delta_F/(B^{1/2}\epsilon^2))$, which implies

$$\mathbb{1}(B < n) \cdot \sigma^2/B \le \epsilon^2/(2C), \quad \text{and} \quad L\Delta_F/(SB^{1/2}) \le \epsilon^2/(2C). \qquad (A.11)$$

Submitting (A.11) into (A.10), we have $\mathbb{E}\|\nabla F(\mathbf{y}_{\text{out}})\|_2^2 \le 2C\epsilon^2/(2C) = \epsilon^2$. By Lemma 4.1, we have that each One-epoch-SNVRG takes less than $7B\log^3 B$ stochastic gradient computations. Since we have total $S$ epochs, so the total gradient complexity of Algorithm 2 is less than

$$S \cdot 7B\log^3 B \le 7B\log^3 B + \frac{L\Delta_F}{\epsilon^2} \cdot 7B^{1/2}\log^3 B$$

$$= O\left(\log^3\left(\frac{\sigma^2}{\epsilon^2} \wedge n\right)\left[\frac{\sigma^2}{\epsilon^2} \wedge n + \frac{L\Delta_F}{\epsilon^2}\left[\frac{\sigma^2}{\epsilon^2} \wedge n\right]^{1/2}\right]\right), \qquad (A.12)$$

which leads to the conclusion. $\square$

## A.3 Proof of Theorem 4.4

We then prove the main theorem on gradient complexity of SNVRG under gradient dominance condition (Algorithm 3).

*Proof of Theorem 4.4.* Following the proof of Theorem 4.2, we obtain a similar inequality with (A.9):

$$\mathbb{E}\|\nabla F(\mathbf{z}_{u+1})\|_2^2 \le C\left(\frac{L\mathbb{E}[F(\mathbf{z}_u) - F^*]}{SB^{1/2}} + \frac{\sigma^2}{B} \cdot \mathbb{1}(B < n)\right). \qquad (A.13)$$

Since $F$ is a $\tau$-gradient dominated function, we have $\mathbb{E}\|\nabla F(\mathbf{z}_{u+1})\|_2^2 \ge 1/\tau \cdot \mathbb{E}[F(\mathbf{z}_{u+1}) - F^*]$ by Definition 2.6. Plugging this inequality into (A.13) yields

$$\mathbb{E}\big[F(\mathbf{z}_{u+1}) - F^*\big] \le \frac{C\tau L}{SB^{1/2}} \cdot \mathbb{E}\big[F(\mathbf{z}_u) - F^*\big] + \frac{C\tau\sigma^2}{B} \cdot \mathbb{1}(B < n)$$

$$\le \frac{1}{2}\mathbb{E}\big[F(\mathbf{z}_u) - F^*\big] + \frac{\epsilon}{4}, \qquad (A.14)$$

where the second inequality holds due to the choice of parameters $B = n \wedge (4C_1\tau\sigma^2/\epsilon)$ and $S = 1 \vee (2C_1\tau L/B^{1/2})$ for Algorithm 3 in Theorem 4.4. By (A.14) we can derive

$$\mathbb{E}\big[F(\mathbf{z}_{u+1}) - F^*\big] - \frac{\epsilon}{2} \le \frac{1}{2}\left(\mathbb{E}\big[F(\mathbf{z}_u) - F^*\big] - \frac{\epsilon}{2}\right),$$

which immediately implies

$$\mathbb{E}\big[F(\mathbf{z}_U) - F^*\big] - \frac{\epsilon}{2} \leq \frac{1}{2^U}\left(\Delta_F - \frac{\epsilon}{2}\right) \leq \frac{\Delta_F}{2^U}. \tag{A.15}$$

Plugging the number of epochs $U = \log(2\Delta_F/\epsilon)$ into (A.15), we obtain $\mathbb{E}\big[F(\mathbf{z}_U) - F^*\big] \leq \epsilon$. Note that each epoch of Algorithm 3 needs at most $S \cdot 7B \log^3 B$ stochastic gradient computations by Theorem 4.2 and Algorithm 3 has $U$ epochs, which implies the total stochastic gradient complexity

$$U \cdot S \cdot 7B \log^3 B = O\left(\log^3\left(n \wedge \frac{\tau\sigma^2}{\epsilon}\right)\log\frac{\Delta_F}{\epsilon}\left[n \wedge \frac{\tau\sigma^2}{\epsilon} + \tau L\left[n \wedge \frac{\tau\sigma^2}{\epsilon}\right]^{1/2}\right]\right). \tag{A.16}$$

$\square$

# B  Proof of Key Lemma A.1

In this section, we focus on proving Lemma A.1 which plays a pivotal role in proving our main theorems. Let $M, \{T_i\}, \{B_i\}, B$ be the parameters as defined in Algorithm 1. We denote $T = \prod_{l=1}^K T_l$. We define filtration $\mathcal{F}_t = \sigma(\mathbf{x}_0, \ldots, \mathbf{x}_t)$. Let $\{\mathbf{x}_t^{(l)}\}, \{\mathbf{g}_t^{(l)}\}$ be the reference points and reference gradients in Algorithm 1. We define $\mathbf{v}_t^{(l)}$ as

$$\mathbf{v}_t^{(l)} := \sum_{j=0}^l \mathbf{g}_t^{(j)}, \quad \text{for } 0 \leq l \leq K. \tag{B.1}$$

We first present the following definition and two technical lemmas for the purpose of our analysis.

**Definition B.1.** We define constant series $\{c_j^{(s)}\}$ as the following. For each $s$, we define $c_{T_s}^{(s)}$ as

$$c_{T_s}^{(s)} = \frac{M}{6^{K-s+1}\prod_{l=s}^K T_l}. \tag{B.2}$$

When $0 \leq j < T_s$, we define $c_j^{(s)}$ by induction:

$$c_j^{(s)} = \left(1 + \frac{1}{T_s}\right)c_{j+1}^{(s)} + \frac{3L^2}{M} \cdot \frac{\prod_{l=s+1}^K T_l}{B_s}. \tag{B.3}$$

**Lemma B.2.** For any $p, s$, where $1 \leq s \leq K$ and $0 \leq p\prod_{j=s}^K T_j < (p+1)\prod_{j=s}^K T_j \leq \prod_{j=1}^K T_j$, we define

$$\text{start} = p \cdot \prod_{j=s}^K T_j, \ \text{end} = \text{start} + \prod_{j=s}^K T_j,$$

for simplification. Then we have the following results:

$$\mathbb{E}\left[\sum_{j=\text{start}}^{\text{end}-1} \frac{\|\nabla F(\mathbf{x}_j)\|_2^2}{100M} + F(\mathbf{x}_\text{end}) + c_{T_s}^{(s)} \cdot \|\mathbf{x}_\text{end} - \mathbf{x}_\text{start}\|_2^2 \big| \mathcal{F}_\text{start}\right]$$

$$\leq F(\mathbf{x}_\text{start}) + \frac{2}{M} \cdot \mathbb{E}\big[\|\nabla F(\mathbf{x}_\text{start}) - \mathbf{v}_\text{start}\|_2^2 \big| \mathcal{F}_\text{start}\big] \cdot \prod_{j=s}^K T_j.$$

**Lemma B.3** (Lei et al. [26]). Let $\mathbf{a}_i$ be vectors satisfying $\sum_{i=1}^N \mathbf{a}_i = 0$. Let $\mathcal{J}$ be a uniform random subset of $\{1, \ldots, N\}$ with size $m$, then

$$\mathbb{E}\left\|\frac{1}{m}\sum_{j\in\mathcal{J}}\mathbf{a}_j\right\|_2^2 \leq \frac{\mathbb{1}(|\mathcal{J}| < N)}{mN}\sum_{j=1}^N \|\mathbf{a}_j\|_2^2.$$

*Proof of Lemma A.1.* We have

$$\sum_{j=0}^{T-1} \frac{\mathbb{E}\|\nabla F(\mathbf{x}_j)\|_2^2}{100M} + \mathbb{E}\big[F(\mathbf{x}_T)\big] \le \sum_{j=0}^{T-1} \frac{\mathbb{E}\|\nabla F(\mathbf{x}_j)\|_2^2}{100M} + \mathbb{E}\big[F(\mathbf{x}_T) + c_{T_1}^{(1)} \cdot \|\mathbf{x}_T - \mathbf{x}_0\|_2^2\big]$$

$$\le \mathbb{E}\big[F(\mathbf{x}_0)\big] + \frac{2}{M} \cdot \mathbb{E}\|\nabla F(\mathbf{x}_0) - \mathbf{g}_0\|_2^2 \cdot T, \qquad \text{(B.4)}$$

where the second inequality comes from Lemma B.2 with we take $s = 1, p = 0$. Moreover we have

$$\mathbb{E}\|\nabla F(\mathbf{x}_0) - \mathbf{g}_0\|_2^2 = \mathbb{E}\bigg\|\frac{1}{B}\sum_{i\in I}\big[\nabla f_i(\mathbf{x}_0) - \nabla F(\mathbf{x}_0)\big]\bigg\|_2^2$$

$$\le \mathbb{1}(B < n) \cdot \frac{1}{B} \cdot \frac{1}{n}\sum_{i=1}^{n}\big\|\nabla f_i(\mathbf{x}_0) - \nabla F(\mathbf{x}_0)\big\|_2^2 \qquad \text{(B.5)}$$

$$\le \mathbb{1}(B < n) \cdot \frac{\sigma^2}{B}, \qquad \text{(B.6)}$$

where (B.5) holds because of Lemma B.3. Plug (B.6) into (B.4) and note that we have $M = 6L$, and then we obtain

$$\sum_{j=0}^{T-1} \mathbb{E}\|\nabla F(\mathbf{x}_j)\|_2^2 \le C\bigg(M\mathbb{E}\big[F(\mathbf{x}_0) - F(\mathbf{x}_T)\big] + \frac{2T\sigma^2}{B} \cdot \mathbb{1}(B < n)\bigg), \qquad \text{(B.7)}$$

where $C = 100$. Divide both sides of (B.7) by $T$, then Lemma A.1 holds trivially. $\square$

# C   Proof of Technical Lemmas

In this section, we provide the proofs of technical lemmas used in Appendix B.

## C.1   Proof of Lemma B.2

Let $M, \{T_l\}, \{B_l\}, B$ be the parameters defined in Algorithm 1 and $\{\mathbf{x}_t^{(l)}\}, \{\mathbf{g}_t^{(l)}\}$ be the reference points and reference gradients defined in Algorithm 1. Let $\mathbf{v}_t^{(l)}, \mathcal{F}_t$ be the variables and filtration defined in Appendix B and let $c_j^{(s)}$ be the constant series defined in Definition B.1.

In order to prove Lemma B.2, we will need the following supporting propositions and lemmas. We first state the proposition about the relationship among $\mathbf{x}_t^{(s)}, \mathbf{g}_t^{(s)}$ and $\mathbf{v}_t^{(s)}$:

**Proposition C.1.** Let $\mathbf{v}_t^{(l)}$ be defined as in (B.1). Let $p, s$ satisfy $0 \le p \cdot \prod_{j=s+1}^{K} T_j < (p+1) \cdot \prod_{j=s+1}^{K} T_j < T$. For any $t, t'$ satisfying $p \cdot \prod_{j=s+1}^{K} T_j \le t < t' < (p+1) \cdot \prod_{j=s+1}^{K} T_j$, it holds that

$$\mathbf{x}_t^{(s)} = \mathbf{x}_{t'}^{(s)} = \mathbf{x}_{p\prod_{j=s+1}^{K} T_j}, \qquad \text{(C.1)}$$

$$\mathbf{g}_t^{(s')} = \mathbf{g}_{t'}^{(s')}, \qquad \text{for any } s' \text{ that satisfies } 0 \le s' \le s, \qquad \text{(C.2)}$$

$$\mathbf{v}_t^{(s)} = \mathbf{v}_{t'}^{(s)} = \mathbf{v}_{p\prod_{j=s+1}^{K} T_j}. \qquad \text{(C.3)}$$

The following lemma spells out the relationship between $c_j^{(s-1)}$ and $c_{T_s}^{(s)}$. In a word, $c_j^{(s-1)}$ is about $1 + T_{s-1}$ times less than $c_{T_s}^{(s)}$:

**Lemma C.2.** If $B_s \ge 6^{K-s+1}(\prod_{l=s}^{K} T_l)^2, T_l \ge 1$ and $M \ge 6L$, then it holds that

$$c_j^{(s-1)} \cdot (1 + T_{s-1}) < c_{T_s}^{(s)}, \qquad \text{for } 2 \le s \le K, 0 \le j \le T_{s-1}, \qquad \text{(C.4)}$$

and

$$c_j^{(K)} \cdot (1 + T_K) < M, \qquad \text{for } 0 \le j \le T_K. \qquad \text{(C.5)}$$

Next lemma is a special case of Lemma B.2 with $s = K$:

**Lemma C.3.** Suppose $p$ satisfies $0 \leq pT_K < (p+1)T_K \leq \prod_{i=1}^{K} T_i$. If $M > L$, then we have

$$\mathbb{E}\left[F(\mathbf{x}_{(p+1)\cdot T_K}) + c_{T_K}^{(K)} \cdot \left\|\mathbf{x}_{(p+1)\cdot T_K} - \mathbf{x}_{p\cdot T_K}\right\|_2^2 + \sum_{j=0}^{T_K-1} \frac{\left\|\nabla F(\mathbf{x}_{p\cdot T_K+j})\right\|_2^2}{100M} \middle| \mathcal{F}_{p\cdot T_K}\right]$$

$$\leq F(\mathbf{x}_{p\cdot T_K}) + \frac{2}{M} \cdot \mathbb{E}\left[\left\|\nabla F(\mathbf{x}_{p\cdot T_K}) - \mathbf{v}_{p\cdot T_K}\right\|_2^2 \middle| \mathcal{F}_{p\cdot T_K}\right] \cdot T_K. \tag{C.6}$$

The following lemma provides an upper bound of $\mathbb{E}\left[\left\|\nabla F(\mathbf{x}_t^{(l)}) - \mathbf{v}_t^{(l)}\right\|_2^2\right]$, which plays an important role in our proof of Lemma B.2.

**Lemma C.4.** Let $t^l$ be as defined in (3.1), then we have $\mathbf{x}_t^{(l)} = \mathbf{x}_{t^l}$, and

$$\mathbb{E}\left[\left\|\nabla F(\mathbf{x}_t^{(l)}) - \mathbf{v}_t^{(l)}\right\|_2^2 \middle| \mathcal{F}_{t^l}\right] \leq \frac{L^2}{B_l}\left\|\mathbf{x}_t^{(l)} - \mathbf{x}_t^{(l-1)}\right\|_2^2 + \left\|\nabla F(\mathbf{x}_t^{(l-1)}) - \mathbf{v}_t^{(l-1)}\right\|_2^2. \tag{C.7}$$

*Proof of Lemma B.2.* We use mathematical induction to prove that Lemma B.2 holds for any $1 \leq s \leq K$. When $s = K$, the statement holds because of Lemma C.3. Suppose that for $s+1$, Lemma B.2 holds for any $p'$ which satisfies $0 \leq p' \prod_{j=s+1}^{K} T_j < (p'+1) \prod_{j=s+1}^{K} T_j \leq \prod_{j=1}^{K} T_j$. We need to prove Lemma B.2 still holds for $s$ and $p$, where $p$ satisfies $0 \leq p \prod_{j=s}^{K} T_j < (p+1) \prod_{j=s}^{K} T_j \leq \prod_{j=1}^{K} T_j$. We first define $m = \prod_{j=s+1}^{K} T_j$ for simplification, then we choose $p' = pT_s + u$, and we set indices $\text{start}_u$ and $\text{end}_u$ as

$$\text{start}_u = p' \prod_{j=s+1}^{K} T_j, \qquad \text{end}_u = \text{start}_u + \prod_{j=s+1}^{K} T_j.$$

It can be easily verified that the following relationship also holds:

$$\text{start}_u = \text{start} + um, \qquad \text{end}_u = \text{start} + (u+1)m. \tag{C.8}$$

Based on (C.8), we have

$$\mathbb{E}\left[\sum_{j=\text{start}_u}^{\text{end}_u-1} \frac{\|\nabla F(\mathbf{x}_j)\|_2^2}{100M} + F(\mathbf{x}_{\text{start}+(u+1)m}) + c_{T_{s+1}}^{(s+1)} \cdot \left\|\mathbf{x}_{\text{start}+(u+1)m} - \mathbf{x}_{\text{start}+um}\right\|_2^2 \middle| \mathcal{F}_{\text{start}_u}\right]$$

$$= \mathbb{E}\left[\sum_{j=\text{start}_u}^{\text{end}_u-1} \frac{\|\nabla F(\mathbf{x}_j)\|_2^2}{100M} + F(\mathbf{x}_{\text{end}_u}) + c_{T_{s+1}}^{(s+1)} \cdot \left\|\mathbf{x}_{\text{end}_u} - \mathbf{x}_{\text{start}_u}\right\|_2^2 \middle| \mathcal{F}_{\text{start}_u}\right]$$

$$\leq F(\mathbf{x}_{\text{start}_u}) + \frac{2}{M} \cdot \mathbb{E}\left[\left\|\nabla F(\mathbf{x}_{\text{start}_u}) - \mathbf{v}_{\text{start}_u}\right\|_2^2 \middle| \mathcal{F}_{\text{start}_u}\right] \cdot \prod_{j=s+1}^{K} T_j, \tag{C.9}$$

where the last inequality holds because of the induction hypothesis that Lemma B.2 holds for $s+1$ and $p'$. Note that we have $\mathbf{x}_{\text{start}_u} = \mathbf{x}_{\text{start}+u\cdot m} = \mathbf{x}_{\text{start}_u}^{(s)}$ from Proposition C.1, which implies

$$\mathbb{E}\left[\left\|\nabla F(\mathbf{x}_{\text{start}_u}) - \mathbf{v}_{\text{start}_u}\right\|_2^2 \middle| \mathcal{F}_{\text{start}_u}\right] = \mathbb{E}\left[\left\|\nabla F(\mathbf{x}_{\text{start}_u}^{(s)}) - \mathbf{v}_{\text{start}_u}^{(s)}\right\|_2^2 \middle| \mathcal{F}_{\text{start}_u}\right]$$

$$\leq \frac{L^2}{B_s}\left\|\mathbf{x}_{\text{start}_u}^{(s)} - \mathbf{x}_{\text{start}_u}^{(s-1)}\right\|_2^2 + \left\|\nabla F(\mathbf{x}_{\text{start}_u}^{(s-1)}) - \mathbf{v}_{\text{start}_u}^{(s-1)}\right\|_2^2 \tag{C.10}$$

$$= \frac{L^2}{B_s}\left\|\mathbf{x}_{\text{start}+u\cdot m} - \mathbf{x}_{\text{start}}\right\|_2^2 + \left\|\nabla F(\mathbf{x}_{\text{start}}) - \mathbf{v}_{\text{start}}\right\|_2^2, \tag{C.11}$$

where (C.10) holds because of Lemma C.4 and (C.11) holds due to Proposition C.1. Plugging (C.11) into (C.9) and taking expectation $\mathbb{E}[\cdot|\mathcal{F}_{\text{start}}]$ for (C.9), we have

$$\mathbb{E}\left[\sum_{j=\text{start}_u}^{\text{end}_u-1} \frac{\|\nabla F(\mathbf{x}_j)\|_2^2}{100M} + F(\mathbf{x}_{\text{start}+(u+1)m}) + c_{T_{s+1}}^{(s+1)}\left\|\mathbf{x}_{\text{start}+(u+1)m} - \mathbf{x}_{\text{start}+um}\right\|_2^2 \middle| \mathcal{F}_{\text{start}}\right]$$

$$\leq \mathbb{E}\left[F(\mathbf{x}_{\text{start}+um}) + \frac{2L^2}{MB_s}\left\|\mathbf{x}_{\text{start}+um} - \mathbf{x}_{\text{start}}\right\|_2^2 \cdot \prod_{j=s+1}^{K} T_j + \frac{2}{M}\left\|\nabla F(\mathbf{x}_{\text{start}}) - \mathbf{v}_{\text{start}}\right\|_2^2 \cdot \prod_{j=s+1}^{K} T_j \middle| \mathcal{F}_{\text{start}}\right]. \tag{C.12}$$

Next we bound $\|\mathbf{x}_{\text{start}+(u+1)\cdot m} - \mathbf{x}_{\text{start}}\|_2^2$ as the following:

$$
\begin{aligned}
&\|\mathbf{x}_{\text{start}+(u+1)\cdot m} - \mathbf{x}_{\text{start}}\|_2^2 \\
&= \|\mathbf{x}_{\text{start}+u\cdot m} - \mathbf{x}_{\text{start}}\|_2^2 + \|\mathbf{x}_{\text{start}+(u+1)\cdot m} - \mathbf{x}_{\text{start}+u\cdot m}\|_2^2 \\
&\quad + 2\langle \mathbf{x}_{\text{start}+(u+1)\cdot m} - \mathbf{x}_{\text{start}+u\cdot m}, \mathbf{x}_{\text{start}+u\cdot m} - \mathbf{x}_{\text{start}}\rangle \\
&\le \|\mathbf{x}_{\text{start}+u\cdot m} - \mathbf{x}_{\text{start}}\|_2^2 + \|\mathbf{x}_{\text{start}+(u+1)\cdot m} - \mathbf{x}_{\text{start}+u\cdot m}\|_2^2 \\
&\quad + \frac{1}{T_s}\cdot\|\mathbf{x}_{\text{start}+u\cdot m} - \mathbf{x}_{\text{start}}\|_2^2 + T_s\cdot\|\mathbf{x}_{\text{start}+(u+1)\cdot m} - \mathbf{x}_{\text{start}+u\cdot m}\|_2^2 \qquad \text{(C.13)}\\
&= \left(1+\frac{1}{T_s}\right)\cdot\|\mathbf{x}_{\text{start}+u\cdot m} - \mathbf{x}_{\text{start}}\|_2^2 + (1+T_s)\cdot\|\mathbf{x}_{\text{start}+(u+1)\cdot m} - \mathbf{x}_{\text{start}+u\cdot m}\|_2^2, \qquad \text{(C.14)}
\end{aligned}
$$

where (C.13) holds because of Young's inequality. Taking expectation $\mathbb{E}[\cdot|\mathcal{F}_{\text{start}}]$ over (C.14) and multiplying $c_{u+1}^{(s)}$ on both sides, we obtain

$$
\begin{aligned}
c_{u+1}^{(s)}\mathbb{E}\big[\|\mathbf{x}_{\text{start}+(u+1)\cdot m} - \mathbf{x}_{\text{start}}\|_2^2\big|\mathcal{F}_{\text{start}}\big] &\le c_{u+1}^{(s)}\left(1+\frac{1}{T_s}\right)\mathbb{E}\big[\|\mathbf{x}_{\text{start}+u\cdot m} - \mathbf{x}_{\text{start}}\|_2^2\big|\mathcal{F}_{\text{start}}\big] \\
&\quad + c_{u+1}^{(s)}(1+T_s)\mathbb{E}\big[\|\mathbf{x}_{\text{start}+(u+1)m} - \mathbf{x}_{\text{start}+um}\|_2^2\big|\mathcal{F}_{\text{start}}\big].
\end{aligned}
$$
$$\text{(C.15)}$$

Adding up inequalities(C.15) and (C.12) together yields

$$
\begin{aligned}
&\mathbb{E}\Bigg[\sum_{j=\text{start}_u}^{\text{end}_u-1}\frac{\|\nabla F(\mathbf{x}_j)\|_2^2}{100M} + F(\mathbf{x}_{\text{start}+(u+1)m}) + c_{u+1}^{(s)}\|\mathbf{x}_{\text{start}+(u+1)m} - \mathbf{x}_{\text{start}}\|_2^2 \\
&\qquad + c_{T_{s+1}}^{(s+1)}\|\mathbf{x}_{\text{start}+(u+1)m} - \mathbf{x}_{\text{start}+um}\|_2^2\Big|\mathcal{F}_{\text{start}}\Bigg] \\
&\le \mathbb{E}\Bigg[F(\mathbf{x}_{\text{start}+um}) + \|\mathbf{x}_{\text{start}+um} - \mathbf{x}_{\text{start}}\|_2^2\bigg[c_{u+1}^{(s)}\left(1+\frac{1}{T_s}\right) + \frac{3L^2}{B_sM}\prod_{j=s+1}^{K}T_j\bigg]\Big|\mathcal{F}_{\text{start}}\Bigg] \\
&\quad + \frac{2}{M}\mathbb{E}\big[\|\nabla F(\mathbf{x}_{\text{start}}) - \mathbf{v}_{\text{start}}\|_2^2\big|\mathcal{F}_{\text{start}}\big]\prod_{j=s+1}^{K}T_j \\
&\quad + c_{u+1}^{(s)}(1+T_s)\mathbb{E}\big[\|\mathbf{x}_{\text{start}+(u+1)m} - \mathbf{x}_{\text{start}+um}\|_2^2\big|\mathcal{F}_{\text{start}}\big] \\
&< \mathbb{E}\big[F(\mathbf{x}_{\text{start}+um}) + c_u^{(s)}\|\mathbf{x}_{\text{start}+um} - \mathbf{x}_{\text{start}}\|_2^2\big|\mathcal{F}_{\text{start}}\big] + \frac{2}{M}\mathbb{E}\big[\|\nabla F(\mathbf{x}_{\text{start}}) - \mathbf{v}_{\text{start}}\|_2^2\big|\mathcal{F}_{\text{start}}\big]\prod_{j=s+1}^{K}T_j \\
&\quad + c_{T_{s+1}}^{(s+1)}\mathbb{E}\big[\|\mathbf{x}_{\text{start}+(u+1)m} - \mathbf{x}_{\text{start}+um}\|_2^2\big|\mathcal{F}_{\text{start}}\big], \qquad \text{(C.16)}
\end{aligned}
$$

where the last inequality holds due to the fact that $c_u^{(s)} = c_{u+1}^{(s)}(1+1/T_s) + 3L^2/(B_sM)\cdot\prod_{j=s+1}^{K}T_j$ by Definition B.1 and $c_{u+1}^{(s)}\cdot(1+T_s) < c_{T_{s+1}}^{(s+1)}$ by Lemma C.2. Cancelling out the term $c_{T_{s+1}}^{(s+1)}\cdot$ $\mathbb{E}\big[\|\mathbf{x}_{\text{start}+(u+1)\cdot m} - \mathbf{x}_{\text{start}+u\cdot m}\|_2^2\big|\mathcal{F}_{\text{start}}\big]$ from both sides of (C.16), we get

$$
\begin{aligned}
&\sum_{j=\text{start}_u}^{\text{end}_u-1}\mathbb{E}\bigg[\frac{\|\nabla F(\mathbf{x}_j)\|_2^2}{100M}\bigg|\mathcal{F}_{\text{start}}\bigg] + \mathbb{E}\big[F(\mathbf{x}_{\text{start}+(u+1)\cdot m}) + c_{u+1}^{(s)}\cdot\|\mathbf{x}_{\text{start}+(u+1)\cdot m} - \mathbf{x}_{\text{start}}\|_2^2\big|\mathcal{F}_{\text{start}}\big] \\
&\le \mathbb{E}\big[F(\mathbf{x}_{\text{start}+um}) + c_u^{(s)}\|\mathbf{x}_{\text{start}+um} - \mathbf{x}_{\text{start}}\|_2^2\big|\mathcal{F}_{\text{start}}\big] \\
&\quad + \frac{2}{M}\mathbb{E}\big[\|\nabla F(\mathbf{x}_{\text{start}}) - \mathbf{v}_{\text{start}}\|_2^2\big|\mathcal{F}_{\text{start}}\big]\prod_{j=s+1}^{K}T_j.
\end{aligned}
$$

We now telescope the above inequality for $u = 0$ to $T_s - 1$, then we have

$$\mathbb{E}\left[\sum_{u=0}^{T_s-1}\sum_{j=\text{start}_u}^{\text{end}_u-1}\frac{\|\nabla F(\mathbf{x}_j)\|_2^2}{100M} + F(\mathbf{x}_{\text{end}}) + c_{T_s}^{(s)}\cdot\|\mathbf{x}_{\text{end}} - \mathbf{x}_{\text{start}}\|_2^2\Big|\mathcal{F}_{\text{start}}\right]$$

$$\leq F(\mathbf{x}_{\text{start}}) + \frac{2T_s}{M}\cdot\mathbb{E}\left[\|\nabla F(\mathbf{x}_{\text{start}}) - \mathbf{v}_{\text{start}}\|_2^2\big|\mathcal{F}_{\text{start}}\right]\cdot\prod_{j=s+1}^{K}T_j.$$

Since $\text{start}_u = \text{end}_{u-1}$, $\text{start}_0 = \text{start}$, and $\text{end}_{T_s-1} = \text{end}$, we have

$$\mathbb{E}\left[\sum_{j=\text{start}}^{\text{end}-1}\frac{\|\nabla F(\mathbf{x}_j)\|_2^2}{100M} + F(\mathbf{x}_{\text{end}}) + c_{T_s}^{(s)}\cdot\|\mathbf{x}_{\text{end}} - \mathbf{x}_{\text{start}}\|_2^2\Big|\mathcal{F}_{\text{start}}\right]$$

$$\leq F(\mathbf{x}_{\text{start}}) + \frac{2}{M}\cdot\mathbb{E}\left[\|\nabla F(\mathbf{x}_{\text{start}}) - \mathbf{v}_{\text{start}}\|_2^2\big|\mathcal{F}_{\text{start}}\right]\cdot\prod_{j=s}^{K}T_j. \tag{C.17}$$

Therefore, we have proved that Lemma B.2 still holds for $s$ and $p$. Then by mathematical induction, we have for all $1 \leq s \leq K$ and $p$ which satisfy $0 \leq p \cdot \prod_{j=s}^{K} T_j < (p+1) \cdot \prod_{j=s}^{K} T_j \leq \prod_{j=1}^{K} T_j$, Lemma B.2 holds. $\qquad\square$

## C.2 Proof of Lemma B.3

The following proof is adapted from that of Lemma A.1 in Lei et al. [26]. We provide the proof here for the self-containedness of our paper.

*Proof of Lemma B.3.* We only consider the case when $m < N$. Let $W_j = \mathbb{1}(j \in \mathcal{J})$, then we have

$$\mathbb{E}W_j^2 = \mathbb{E}W_j = \frac{m}{N}, \mathbb{E}W_jW_{j'} = \frac{m(m-1)}{N(N-1)}. \tag{C.18}$$

Thus we can rewrite the sample mean as

$$\frac{1}{m}\sum_{j\in\mathcal{J}}\mathbf{a}_j = \frac{1}{m}\sum_{i=1}^{N}W_i\mathbf{a}_i, \tag{C.19}$$

which immediately implies

$$\mathbb{E}\left\|\frac{1}{m}\sum_{j\in\mathcal{J}}\mathbf{a}_j\right\|^2 = \frac{1}{m^2}\left(\sum_{j=1}^{N}\mathbb{E}W_j^2\|\mathbf{a}_j\|_2^2 + \sum_{j\neq j'}\mathbb{E}W_jW_{j'}\langle\mathbf{a}_j,\mathbf{a}_{j'}\rangle\right)$$

$$= \frac{1}{m^2}\left(\frac{m}{N}\sum_{j=1}^{N}\|\mathbf{a}_j\|_2^2 + \frac{m(m-1)}{N(N-1)}\sum_{j\neq j'}\langle\mathbf{a}_j,\mathbf{a}_{j'}\rangle\right)$$

$$= \frac{1}{m^2}\left(\left(\frac{m}{N} - \frac{m(m-1)}{N(N-1)}\right)\sum_{j=1}^{N}\|\mathbf{a}_j\|_2^2 + \frac{m(m-1)}{N(N-1)}\left\|\sum_{j=1}^{N}\mathbf{a}_j\right\|_2^2\right)$$

$$= \frac{1}{m^2}\left(\frac{m}{N} - \frac{m(m-1)}{N(N-1)}\right)\sum_{j=1}^{N}\|\mathbf{a}_j\|_2^2$$

$$\leq \frac{1}{m}\cdot\frac{1}{N}\sum_{j=1}^{N}\|\mathbf{a}_j\|_2^2.$$

$\qquad\square$

# D  Proofs of the Auxiliary Lemmas

In this section, we present the additional proofs of supporting lemmas used in Appendix C. Let $M, \{T_l\}, \{B_l\}$ and $B$ be the parameters defined in Algorithm 1. Let $\{\mathbf{x}_t^{(l)}\}, \{\mathbf{g}_t^{(l)}\}$ be the reference points and reference gradients used in Algorithm 1. Finally, $\mathbf{v}_t^{(l)}, \mathcal{F}_t$ are the variables and filtration defined in Appendix B and $c_j^{(s)}$ are the constant series defined in Definition B.1.

## D.1  Proof of Proposition C.1

*Proof of Proposition C.1.* By the definition of reference point $\mathbf{x}_t^{(s)}$ in (3.1), we can easily verify that (C.1) holds trivially.

Next we prove (C.2). Note that by (C.1) we have $\mathbf{x}_t^{(s)} = \mathbf{x}_{t'}^{(s)}$. For any $0 \le s' \le s$, it is also true that $\mathbf{x}_t^{(s')} = \mathbf{x}_{t'}^{(s')}$ by (3.1), which means $\mathbf{x}_t$ and $\mathbf{x}_{t'}$ share the same first $s+1$ reference points. Then by the update rule of $\mathbf{g}_t^{(s')}$ in Algorithm 1, we will maintain $\mathbf{g}_t^{(s')}$ unchanged from time step $t$ to $t'$. In other worlds, we have $\mathbf{g}_t^{(s')} = \mathbf{g}_{t'}^{(s')}$ for all $0 \le s' \le s$.

We now prove the last claim (C.3). Based on (B.1) and (C.2), we have $\mathbf{v}_t^{(s)} = \sum_{s'=0}^{s} \mathbf{g}_t^{(s')} = \sum_{s'=0}^{s} \mathbf{g}_{p \cdot \prod_{j=s+1}^{K} T_j}^{(s')} = \mathbf{v}_{p \cdot \prod_{j=s+1}^{K} T_j}^{(s)}$. Since for any $s \le s'' \le K$, we have the following equations by the update in Algorithm 1 (Line 18).

$$
\begin{aligned}
\mathbf{x}_{p \cdot \prod_{j=s+1}^{K} T_j}^{(s'')} &= \mathbf{x}_{\lfloor p \cdot \prod_{j=s+1}^{K} T_j / \prod_{j=s''+1}^{K} T_j \rfloor \cdot \prod_{j=s''+1}^{K} T_j} \\
&= \mathbf{x}_{p \cdot \prod_{j=s+1}^{K} T_j / \prod_{j=s''+1}^{K} T_j \cdot \prod_{j=s''+1}^{K} T_j} \\
&= \mathbf{x}_{p \cdot \prod_{j=s+1}^{K} T_j}^{(s)}.
\end{aligned}
$$

Then for any $s < s'' \le K$, we have

$$
\mathbf{g}_{p \cdot \prod_{j=s+1}^{K} T_j}^{(s'')} = \frac{1}{B_{s''}} \sum_{i \in I} \left[ \nabla f_i\left(\mathbf{x}_{p \cdot \prod_{j=s+1}^{K} T_j}^{(s'')}\right) - \nabla f_i\left(\mathbf{x}_{p \cdot \prod_{j=s+1}^{K} T_j}^{(s''-1)}\right) \right] = 0. \tag{D.1}
$$

Thus, we have

$$
\mathbf{v}_{p \cdot \prod_{j=s+1}^{K} T_j} = \sum_{s''=0}^{K} \mathbf{g}_{p \cdot \prod_{j=s+1}^{K} T_j}^{(s'')} = \sum_{s''=0}^{s} \mathbf{g}_{p \cdot \prod_{j=s+1}^{K} T_j}^{(s'')} = \sum_{s''=0}^{s} \mathbf{g}_t^{(s'')} = \mathbf{v}_t^{(s)}, \tag{D.2}
$$

where the first equality holds because of the definition of $\mathbf{v}_{p \cdot \prod_{j=s+1}^{K} T_j}$, the second equality holds due to (D.1), the third equality holds due to (C.2) and the last equality holds due to (B.1). This completes the proof of (C.3). $\qquad\square$

## D.2  Proof of Lemma C.2

*Proof of Lemma C.2.* For any fixed $s$, it can be seen that from the definition in (B.3), $c_j^{(s)}$ is monotonically decreasing with $j$. In order to prove (C.4), we only need to compare $(1+T_{s-1}) \cdot c_0^{(s-1)}$ and $c_{T_s}^{(s)}$. Furthermore, by the definition of series $\{c_j^{(s)}\}$ in (B.3), it can be inducted that when $0 \le j \le T_{s-1}$,

$$
c_j^{(s-1)} = \left(1 + \frac{1}{T_{s-1}}\right)^{T_{s-1}-j} \cdot c_{T_{s-1}}^{(s-1)} + \frac{(1+1/T_{s-1})^{T_{s-1}-j} - 1}{1/T_{s-1}} \cdot \frac{3L^2}{M} \cdot \frac{\prod_{l=s}^{K} T_l}{B_{s-1}}. \tag{D.3}
$$

We take $j = 0$ in (D.3) and obtain

$$c_0^{(s-1)} = \left(1 + \frac{1}{T_{s-1}}\right)^{T_{s-1}} \cdot c_{T_{s-1}}^{(s-1)} + \frac{(1 + 1/T_{s-1})^{T_{s-1}} - 1}{1/T_{s-1}} \cdot \frac{3L^2}{M} \cdot \frac{\prod_{l=s}^{K} T_l}{B_{s-1}}$$

$$< 2.8 \times c_{T_{s-1}}^{(s-1)} + \frac{6L^2}{M} \cdot \frac{\prod_{l=s-1}^{K} T_l}{B_{s-1}} \tag{D.4}$$

$$\leq \frac{2.8M + 6L^2/M}{6^{K-s+2} \cdot \prod_{l=s-1}^{K} T_l} \tag{D.5}$$

$$< \frac{3M}{6^{K-s+2} \cdot \prod_{l=s-1}^{K} T_l}, \tag{D.6}$$

where (D.4) holds because $(1 + 1/n)^n < 2.8$ for any $n \geq 1$, (D.5) holds due to the definition of $c_{T_{s-1}}^{(s-1)}$ in (B.2) and $B_{s-1} \geq 6^{K-s+2}(\prod_{l=s-1}^{K} T_l)^2$ and (D.6) holds because $M \geq 6L$. Recall that $c_j^{(s)}$ is monotonically decreasing with $j$ and the inequality in (D.6). Thus for all $2 \leq s \leq K$ and $0 \leq j \leq T_{s-1}$, we have

$$(1 + T_{s-1}) \cdot c_j^{(s-1)} \leq (1 + T_{s-1}) \cdot c_0^{(s-1)}$$

$$\leq (1 + T_{s-1}) \cdot \frac{3M}{6^{K-s+2} \cdot \prod_{l=s-1}^{K} T_l}$$

$$< \frac{6M}{6^{K-s+2} \cdot \prod_{l=s}^{K} T_l}$$

$$= c_{T_s}^{(s)}, \tag{D.7}$$

where the third inequality holds because $(1 + T_{s-1})/T_{s-1} \leq 2$ when $T_{s-1} \geq 1$ and the last equation comes from the definition of $c_{T_s}^s$ in (B.2). This completes the proof of (C.4).

Using similar techniques, we can obtain the upper bound for $c_0^K$ which is similar to inequality (D.6) with $s - 1$ replaced by $K$. Therefore, we have

$$(1 + T_K) \cdot c_j^{(K)} \leq (1 + T_K) \cdot c_0^{(K)} < \frac{6M}{6^{K-K+1} \cdot \prod_{l=K}^{K} T_l} \leq M,$$

which completes the proof of (C.5). $\qquad\square$

### D.3  Proof of Lemma C.3

Now we prove Lemma C.3, which is a special case of Lemma B.2 if we choose $s = K$.

*Proof of Lemma C.3.* To simplify notations, we use $\mathbb{E}[\cdot]$ to denote the conditional expectation $\mathbb{E}[\cdot|\mathcal{F}_{p \cdot T_K}]$ in the rest of this proof. For $0 \leq j < T_K$, we denote $\mathbf{h}_{p \cdot T_K + j} = -(10M)^{-1} \cdot \mathbf{v}_{p \cdot T_K + j}$. According to the update in Algorithm 1 (Line 12), we have

$$\mathbf{x}_{p \cdot T_K + j + 1} = \mathbf{x}_{p \cdot T_K + j} + \mathbf{h}_{p \cdot T_K + j}, \tag{D.8}$$

which immediately implies

$$F(\mathbf{x}_{p \cdot T_K + j + 1}) = F(\mathbf{x}_{p \cdot T_K + j} + \mathbf{h}_{p \cdot T_K + j})$$

$$\leq F(\mathbf{x}_{p \cdot T_K + j}) + \langle \nabla F(\mathbf{x}_{p \cdot T_K + j}), \mathbf{h}_{p \cdot T_K + j} \rangle + \frac{L}{2} \|\mathbf{h}_{p \cdot T_K + j}\|_2^2 \tag{D.9}$$

$$= \left[ \langle \mathbf{v}_{p \cdot T_K + j}, \mathbf{h}_{p \cdot T_K + j} \rangle + 5M \|\mathbf{h}_{p \cdot T_K + j}\|_2^2 \right] + F(\mathbf{x}_{p \cdot T_K + j})$$

$$+ \langle \nabla F(\mathbf{x}_{p \cdot T_K + j}) - \mathbf{v}_{p \cdot T_K + j}, \mathbf{h}_{p \cdot T_K + j} \rangle + \left( \frac{L}{2} - 5M \right) \|\mathbf{h}_{p \cdot T_K + j}\|_2^2$$

$$\leq F(\mathbf{x}_{p \cdot T_K + j}) + \langle \nabla F(\mathbf{x}_{p \cdot T_K + j}) - \mathbf{v}_{p \cdot T_K + j}, \mathbf{h}_{p \cdot T_K + j} \rangle + (L - 5M) \|\mathbf{h}_{p \cdot T_K + j}\|_2^2, \tag{D.10}$$

where (D.9) is due to the $L$-smoothness of $F$, which can be verified as follows

$$\|\nabla F(\mathbf{x}) - \nabla F(\mathbf{y})\|_2 = \|\mathbb{E}_i[\nabla f_i(\mathbf{x}) - \nabla f_i(\mathbf{y})]\|_2$$
$$\leq \sqrt{\mathbb{E}_i\|\nabla f_i(\mathbf{x}) - \nabla f_i(\mathbf{y})\|_2^2}$$
$$\leq L\|\mathbf{x} - \mathbf{y}\|_2.$$

(D.10) holds because $\langle \mathbf{v}_{p\cdot T_K+j}, \mathbf{h}_{p\cdot T_K+j}\rangle + 5M\|\mathbf{h}_{p\cdot T_K+j}\|_2^2 = -5M\|\mathbf{h}_{p\cdot T_K+j}\|_2^2 \leq 0$. Further by Young's inequality, we obtain

$$F(\mathbf{x}_{p\cdot T_K+j+1}) \leq F(\mathbf{x}_{p\cdot T_K+j}) + \frac{1}{2M}\|\nabla F(\mathbf{x}_{p\cdot T_K+j}) - \mathbf{v}_{p\cdot T_K+j}\|_2^2 + \left(\frac{M}{2} + L - 5M\right)\|\mathbf{h}_{p\cdot T_K+j}\|_2^2$$

$$\leq F(\mathbf{x}_{p\cdot T_K+j}) + \frac{1}{M}\|\nabla F(\mathbf{x}_{p\cdot T_K+j}) - \mathbf{v}_{p\cdot T_K+j}\|_2^2 - 3M\|\mathbf{h}_{p\cdot T_K+j}\|_2^2, \qquad \text{(D.11)}$$

where the second inequality holds because $M > L$. Now we bound the term $c_{j+1}^{(K)}\|\mathbf{x}_{p\cdot T_K+j+1} - \mathbf{x}_{p\cdot T_K}\|_2^2$. By (D.8) we have

$$c_{j+1}^{(K)}\|\mathbf{x}_{p\cdot T_K+j+1} - \mathbf{x}_{p\cdot T_K}\|_2^2 = c_{j+1}^{(K)}\|\mathbf{x}_{p\cdot T_K+j} - \mathbf{x}_{p\cdot T_K} + \mathbf{h}_{p\cdot T_K+j}\|_2^2$$
$$= c_{j+1}^{(K)}\left[\|\mathbf{x}_{p\cdot T_K+j} - \mathbf{x}_{p\cdot T_K}\|_2^2 + \|\mathbf{h}_{p\cdot T_K+j}\|_2^2 + 2\langle \mathbf{x}_{p\cdot T_K+j} - \mathbf{x}_{p\cdot T_K}, \mathbf{h}_{p\cdot T_K+j}\rangle\right].$$

Applying Young's inequality yields

$$c_{j+1}^{(K)}\|\mathbf{x}_{p\cdot T_K+j+1} - \mathbf{x}_{p\cdot T_K}\|_2^2 \leq c_{j+1}^{(K)}\Bigg[\|\mathbf{x}_{p\cdot T_K+j} - \mathbf{x}_{p\cdot T_K}\|_2^2 + \|\mathbf{h}_{p\cdot T_K+j}\|_2^2$$

$$+ \frac{1}{T_K}\|\mathbf{x}_{p\cdot T_K+j} - \mathbf{x}_{p\cdot T_K}\|_2^2 + T_K\|\mathbf{h}_{p\cdot T_K+j}\|_2^2\Bigg]$$

$$= c_{j+1}^{(K)}\Bigg[\left(1 + \frac{1}{T_K}\right)\|\mathbf{x}_{p\cdot T_K+j} - \mathbf{x}_{p\cdot T_K}\|_2^2 + (1 + T_K)\|\mathbf{h}_{p\cdot T_K+j}\|_2^2\Bigg],$$
$$\text{(D.12)}$$

Adding up inequalities (D.12) and (D.11), we get

$$F(\mathbf{x}_{p\cdot T_K+j+1}) + c_{j+1}^{(K)}\|\mathbf{x}_{p\cdot T_K+j+1} - \mathbf{x}_{p\cdot T_K}\|_2^2$$
$$\leq F(\mathbf{x}_{p\cdot T_K+j}) + \frac{1}{M}\|\nabla F(\mathbf{x}_{p\cdot T_K+j}) - \mathbf{v}_{p\cdot T_K+j}\|_2^2 - \left[3M - c_{j+1}^{(K)}(1 + T_K)\right]\|\mathbf{h}_{p\cdot T_K+j}\|_2^2$$
$$+ c_{j+1}^{(K)}\left(1 + \frac{1}{T_K}\right)\|\mathbf{x}_{p\cdot T_K+j} - \mathbf{x}_{p\cdot T_K}\|_2^2$$
$$\leq F(\mathbf{x}_{p\cdot T_K+j}) + \frac{1}{M}\|\nabla F(\mathbf{x}_{p\cdot T_K+j}) - \mathbf{v}_{p\cdot T_K+j}\|_2^2 - 2M\|\mathbf{h}_{p\cdot T_K+j}\|_2^2$$
$$+ c_{j+1}^{(K)}\left(1 + \frac{1}{T_K}\right)\|\mathbf{x}_{p\cdot T_K+j} - \mathbf{x}_{p\cdot T_K}\|_2^2, \qquad \text{(D.13)}$$

where the last inequality holds due to the fact that $c_{j+1}^{(K)}(1 + T_K) < M$ by Lemma C.2. Next we bound $\|\nabla F(\mathbf{x}_{p\cdot T_K+j})\|_2^2$ with $\|\mathbf{h}_{p\cdot T_K+j}\|_2^2$. Note that by (D.8), we have

$$\|\nabla F(\mathbf{x}_{p\cdot T_K+j})\|_2^2 = \left\|\left[\nabla F(\mathbf{x}_{p\cdot T_K+j}) - \mathbf{v}_{p\cdot T_K+j}\right] - 10M\mathbf{h}_{p\cdot T_K+j}\right\|_2^2$$
$$\leq 2\left(\|\nabla F(\mathbf{x}_{p\cdot T_K+j}) - \mathbf{v}_{p\cdot T_K+j}\|_2^2 + 100M^2\|\mathbf{h}_{p\cdot T_K+j}\|_2^2\right),$$

which immediately implies

$$-2M\|\mathbf{h}_{p\cdot T_K+j}\|_2^2 \leq \frac{2}{100M}\left(\|\nabla F(\mathbf{x}_{p\cdot T_K+j}) - \mathbf{v}_{p\cdot T_K+j}\|_2^2 - \frac{1}{100M}\|\nabla F(\mathbf{x}_{p\cdot T_K+j})\|_2^2. \quad \text{(D.14)}$$

Plugging (D.14) into (D.13), we have

$$F(\mathbf{x}_{p \cdot T_K + j + 1}) + c_{j+1}^{(K)} \|\mathbf{x}_{p \cdot T_K + j + 1} - \mathbf{x}_{p \cdot T_K}\|_2^2$$

$$\leq F(\mathbf{x}_{p \cdot T_K + j}) + \frac{1}{M} \|\nabla F(\mathbf{x}_{p \cdot T_K + j}) - \mathbf{v}_{p \cdot T_K + j}\|_2^2 + \frac{1}{50M} \cdot \|\nabla F(\mathbf{x}_{p \cdot T_K + j}) - \mathbf{v}_{p \cdot T_K + j}\|_2^2$$

$$- \frac{1}{100M} \|\nabla F(\mathbf{x}_{p \cdot T_K + j})\|_2^2 + c_{j+1}^{(K)} \left(1 + \frac{1}{T_K}\right) \|\mathbf{x}_{p \cdot T_K + j} - \mathbf{x}_{p \cdot T_K}\|_2^2$$

$$\leq F(\mathbf{x}_{p \cdot T_K + j}) + \frac{2}{M} \|\nabla F(\mathbf{x}_{p \cdot T_K + j}) - \mathbf{v}_{p \cdot T_K + j}\|_2^2 - \frac{1}{100M} \|\nabla F(\mathbf{x}_{p \cdot T_K + j})\|_2^2$$

$$+ c_{j+1}^{(K)} \left(1 + \frac{1}{T_K}\right) \|\mathbf{x}_{p \cdot T_K + j} - \mathbf{x}_{p \cdot T_K}\|_2^2. \tag{D.15}$$

Next we bound $\|\nabla F(\mathbf{x}_{p \cdot T_K + j}) - \mathbf{v}_{p \cdot T_K + j}\|_2^2$. First, by Lemma C.4 we have

$$\mathbb{E}\left\|\nabla F(\mathbf{x}_{p \cdot T_K + j}^{(K)}) - \mathbf{v}_{p \cdot T_K + j}^{(K)}\right\|_2^2 \leq \frac{L^2}{B_K} \mathbb{E}\left\|\mathbf{x}_{p \cdot T_K + j}^{(K)} - \mathbf{x}_{p \cdot T_K + j}^{(K-1)}\right\|_2^2 + \mathbb{E}\left\|\nabla F(\mathbf{x}_{p \cdot T_K + j}^{(K-1)}) - \mathbf{v}_{p \cdot T_K + j}^{(K-1)}\right\|_2^2.$$

Since $\mathbf{x}_{p \cdot T_K + j}^{(K)} = \mathbf{x}_{p \cdot T_K + j}, \mathbf{v}_{p \cdot T_K + j}^{(K)} = \mathbf{v}_{p \cdot T_K + j}, \mathbf{x}_{p \cdot T_K + j}^{(K-1)} = \mathbf{x}_{p \cdot T_K}$ and $\mathbf{v}_{p \cdot T_K + j}^{(K-1)} = \mathbf{v}_{p \cdot T_K}$, we have

$$\mathbb{E}\|\nabla F(\mathbf{x}_{p \cdot T_K + j}) - \mathbf{v}_{p \cdot T_K + j}\|_2^2 \leq \frac{L^2}{B_K} \mathbb{E}\|\mathbf{x}_{p \cdot T_K + j} - \mathbf{x}_{p \cdot T_K}\|_2^2 + \mathbb{E}\|\nabla F(\mathbf{x}_{p \cdot T_K}) - \mathbf{v}_{p \cdot T_K}\|_2^2. \tag{D.16}$$

We now take expectation $\mathbb{E}[\cdot]$ with (D.15) and plug (D.16) into (D.15). We obtain that

$$\mathbb{E}\left[F(\mathbf{x}_{p \cdot T_K + j + 1}) + c_{j+1}^{(K)} \|\mathbf{x}_{p \cdot T_K + j + 1} - \mathbf{x}_{p \cdot T_K}\|_2^2 + \frac{1}{100M} \|\nabla F(\mathbf{x}_{p \cdot T_K + j})\|_2^2\right]$$

$$\leq \mathbb{E}\left[F(\mathbf{x}_{p \cdot T_K + j}) + \left(c_{j+1}^{(K)}\left(1 + \frac{1}{T_K}\right) + \frac{3L^2}{B_K M}\right) \|\mathbf{x}_{p \cdot T_K + j} - \mathbf{x}_{p \cdot T_K}\|_2^2 + \frac{2}{M} \|\nabla F(\mathbf{x}_{p \cdot T_K}) - \mathbf{v}_{p \cdot T_K}\|_2^2\right]$$

$$= \mathbb{E}\left[F(\mathbf{x}_{p \cdot T_K + j}) + c_j^{(K)} \|\mathbf{x}_{p \cdot T_K + j} - \mathbf{x}_{p \cdot T_K}\|_2^2 + \frac{2}{M} \cdot \|\nabla F(\mathbf{x}_{p \cdot T_K}) - \mathbf{v}_{p \cdot T_K}\|_2^2\right], \tag{D.17}$$

where (D.17) holds because we have $c_j^{(K)} = c_{j+1}^{(K)}(1 + 1/T_K) + 3L^2/(B_K M)$ by Definition B.1. Telescoping (D.17) for $j = 0$ to $T_K - 1$, we have

$$\mathbb{E}\left[F(\mathbf{x}_{(p+1) \cdot T_K}) + c_{T_K}^{(K)} \cdot \|\mathbf{x}_{(p+1) \cdot T_K} - \mathbf{x}_{p \cdot T_K}\|_2^2\right] + \frac{1}{100M} \sum_{j=0}^{T_K - 1} \mathbb{E}\|\nabla F(\mathbf{x}_{p \cdot T_K + j})\|_2^2$$

$$\leq F(\mathbf{x}_{p \cdot T_K}) + \frac{2T_K}{M} \cdot \mathbb{E}\|\nabla F(\mathbf{x}_{p \cdot T_K}) - \mathbf{v}_{p \cdot T_K}\|_2^2, \tag{D.18}$$

which completes the proof.

$\square$

## D.4 Proof of Lemma C.4

*Proof of Lemma C.4.* If $t^l = t^{l-1}$, we have $\mathbf{x}_t^{(l)} = \mathbf{x}_t^{(l-1)}$ and $\mathbf{v}_t^{(l)} = \mathbf{v}_t^{(l-1)}$. In this case the statement in Lemma C.4 holds trivially. Therefore, we assume $t^l \neq t^{l-1}$ in the following proof. Note that

$$\mathbb{E}\left[\|\nabla F(\mathbf{x}_t^{(l)}) - \mathbf{v}_t^{(l)}\|_2^2 | \mathcal{F}_{t^l}\right]$$

$$= \mathbb{E}\left[\|\nabla F(\mathbf{x}_t^{(l)}) - \mathbf{v}_t^{(l)} - \mathbb{E}[\nabla F(\mathbf{x}_t^{(l)}) - \mathbf{v}_t^{(l)}]\|_2^2 | \mathcal{F}_{t^l}\right] + \|\mathbb{E}[\nabla F(\mathbf{x}_t^{(l)}) - \mathbf{v}_t^{(l)} | \mathcal{F}_{t^l}]\|_2^2$$

$$= \underbrace{\mathbb{E}\left[\left\|\nabla F(\mathbf{x}_t^{(l)}) - \sum_{j=0}^l \mathbf{g}_t^{(j)} - \mathbb{E}\left[\nabla F(\mathbf{x}_t^{(l)}) - \sum_{j=0}^l \mathbf{g}_t^{(j)}\right]\right\|_2^2 \Big| \mathcal{F}_{t^l}\right]}_{J_1} + \underbrace{\left\|\mathbb{E}\left[\nabla F(\mathbf{x}_t^{(l)}) - \sum_{j=0}^l \mathbf{g}_t^{(j)} \Big| \mathcal{F}_{t^l}\right]\right\|_2^2}_{J_2},$$

$$\tag{D.19}$$

where in the second equation we used the definition $\mathbf{v}_t^{(l)} = \sum_{i=0}^{l} \mathbf{g}_t^{(i)}$ in (B.1). We first upper bound term $J_1$. According to the update rule in Algorithm 1 (Line 21-25), when $j < l$, $\mathbf{g}_t^{(j)}$ will not be updated at the $t^l$-th iteration. Thus we have $\mathbb{E}[\mathbf{g}_t^{(j)}|\mathcal{F}_{t^l}] = \mathbf{g}_t^{(j)}$ for all $j < l$. In addition, by the definition of $\mathcal{F}_{t^l}$, we have $\mathbb{E}[\nabla F(\mathbf{x}_t^{(l)})|\mathcal{F}_{t^l}] = \nabla F(\mathbf{x}_t^{(l)})$. Then we have the following equation

$$J_1 = \mathbb{E}\big[\big\|\mathbf{g}_t^{(l)} - \mathbb{E}\big[\mathbf{g}_t^{(l)}|\mathcal{F}_{t^l}\big]\big\|_2^2 \big| \mathcal{F}_{t^l}\big]. \tag{D.20}$$

We further have

$$\mathbf{g}_t^{(l)} = \frac{1}{B_l} \sum_{i \in I} \big[\nabla f_i(\mathbf{x}_t^{(l)}) - \nabla f_i(\mathbf{x}_t^{(l-1)})\big], \quad \mathbb{E}\big[\mathbf{g}_t^{(l)}|\mathcal{F}_{t^l}\big] = \nabla F(\mathbf{x}_t^{(l)}) - \nabla F(\mathbf{x}_t^{(l-1)}). \tag{D.21}$$

Therefore, we can apply Lemma B.3 to (D.20) and obtain

$$\begin{aligned}
J_1 &\leq \frac{1}{B_l} \cdot \frac{1}{n} \sum_{i=1}^{n} \big\|\nabla f_i(\mathbf{x}_t^{(l)}) - \nabla f_i(\mathbf{x}_t^{(l-1)}) - \big[\nabla F(\mathbf{x}_t^{(l)}) - \nabla F(\mathbf{x}_t^{(l-1)})\big]\big\|_2^2 \\
&\leq \frac{1}{B_l n} \sum_{i=1}^{n} \big\|\nabla f_i(\mathbf{x}_t^{(l)}) - \nabla f_i(\mathbf{x}_t^{(l-1)})\big\|_2^2 \\
&\leq \frac{L^2}{B_l} \big\|\mathbf{x}_t^{(l)} - \mathbf{x}_t^{(l-1)}\big\|_2^2,
\end{aligned} \tag{D.22}$$

where the second inequality is due to the fact that $\mathbb{E}[\|\mathbf{X} - \mathbb{E}[\mathbf{X}]\|_2^2] \leq \mathbb{E}\|\mathbf{X}\|_2^2$ for any random vector $\mathbf{X}$ and the last inequality holds due to the fact that $F$ has averaged $L$-Lipschitz gradient.

Next we turn to bound term $J_2$. Note that

$$\mathbb{E}\big[\mathbf{g}_t^{(l)}|\mathcal{F}_{t^l}\big] = \mathbb{E}\bigg[\frac{1}{B_l} \sum_{i \in I} \big[\nabla f_i(\mathbf{x}_t^{(l)}) - \nabla f_i(\mathbf{x}_t^{(l-1)})\big] \bigg| \mathcal{F}_{t^l}\bigg] = \nabla F(\mathbf{x}_t^{(l)}) - \nabla F(\mathbf{x}_t^{(l-1)}),$$

which immediately implies

$$\begin{aligned}
\mathbb{E}\bigg[\nabla F(\mathbf{x}_t^{(l)}) - \sum_{j=0}^{l} \mathbf{g}_t^{(j)} \bigg| \mathcal{F}_{t^l}\bigg] &= \mathbb{E}\bigg[\nabla F(\mathbf{x}_t^{(l)}) - \nabla F(\mathbf{x}_t^{(l)}) + \nabla F(\mathbf{x}_t^{(l-1)}) - \sum_{j=0}^{l-1} \mathbf{g}_t^{(j)} \bigg| \mathcal{F}_{t^l}\bigg] \\
&= \mathbb{E}\big[\nabla F(\mathbf{x}_t^{(l-1)}) - \mathbf{v}_t^{(l-1)} \big| \mathcal{F}_{t^l}\big] \\
&= \nabla F(\mathbf{x}_t^{(l-1)}) - \mathbf{v}_t^{(l-1)},
\end{aligned} \tag{D.23}$$

where the last equation is due to the definition of $\mathcal{F}_t$. Plugging $J_1$ and $J_2$ into (D.19) yields the following result:

$$\mathbb{E}\big[\big\|\nabla F(\mathbf{x}_t^{(l)}) - \mathbf{v}_t^{(l)}\big\|_2^2 \big| \mathcal{F}_{t^l}\big] \leq \frac{L^2}{B_l} \big\|\mathbf{x}_t^{(l)} - \mathbf{x}_t^{(l-1)}\big\|_2^2 + \big\|\nabla F(\mathbf{x}_t^{(l-1)}) - \mathbf{v}_t^{(l-1)}\big\|_2^2, \tag{D.24}$$

which completes the proof. $\qquad\square$

# E   Additional Experimental Results

We also conducted experiments comparing different algorithms without the learning rate decay schedule. The parameters are tuned by the same grid search described in Section 5. In particular, we summarize the parameters of different algorithms used in our experiments with and without learning rate decay for MNIST in Table 2, CIFAR10 in Table 3, and SVHN in Table 4. We plotted the training loss and test error for each dataset without learning rate decay in Figure 4. The results on MNIST are presented in Figures 4(a) and 4(d); the results on CIFAR10 are in Figures 4(b) and 4(e); and the results on SVHN dataset are shown in Figures 4(c) and 4(f). It can be seen that without learning decay, our algorithm *SNVRG* still outperforms all the baseline algorithms except for the training loss on SVHN dataset. However, *SNVRG* still performs the best in terms of test error on SVHN dataset. These results again suggest that *SNVRG* can beat the state-of-the-art in practice, which backups our theory.

Table 2: Parameter settings of all algorithms on MNIST dataset.

| Algorithm | With Learning Rate Decay | | | Without Learning Rate Decay | | |
|---|---|---|---|---|---|---|
| | Initial learning rate $\eta$ | Batch size $B$ | Batch size ratio $b$ | learning rate $\eta$ | Batch size $B$ | Batch size ratio $b$ |
| SGD | 0.1 | 1024 | N/A | 0.01 | 1024 | N/A |
| SGD-momentum | 0.01 | 1024 | N/A | 0.1 | 1024 | N/A |
| ADAM | 0.001 | 1024 | N/A | 0.001 | 1024 | N/A |
| SCSG | 0.01 | 512 | 8 | 0.01 | 512 | 8 |
| SNVRG | 0.01 | 512 | 8 | 0.01 | 512 | 8 |

Table 3: Parameter settings of all algorithms on CIFAR10 dataset.

| Algorithm | With Learning Rate Decay | | | Without Learning Rate Decay | | |
|---|---|---|---|---|---|---|
| | Initial learning rate $\eta$ | Batch size $B$ | Batch size ratio $b$ | learning rate $\eta$ | Batch size $B$ | Batch size ratio $b$ |
| SGD | 0.1 | 1024 | N/A | 0.01 | 512 | N/A |
| SGD-momentum | 0.01 | 1024 | N/A | 0.01 | 2048 | N/A |
| ADAM | 0.001 | 1024 | N/A | 0.001 | 2048 | N/A |
| SCSG | 0.01 | 512 | 8 | 0.01 | 512 | 8 |
| SNVRG | 0.01 | 1024 | 8 | 0.01 | 512 | 4 |

(a) training loss (*MNIST*)    (b) training loss (*CIFAR10*)    (c) training loss (*SVHN*)

(d) test error (*MNIST*)    (e) test error (*CIFAR10*)    (f) test error (*SVHN*)

Figure 4: Experimental results on different datasets without learning rate decay. (a) and (d) depict the training loss and test error (top-1 error) v.s. data epochs for training LeNet on MNIST dataset. (b) and (e) depict the training loss and test error v.s. data epochs for training LeNet on CIFAR10 dataset. (c) and (f) depict the training loss and test error v.s. data epochs for training LeNet on SVHN dataset.

# F    An Equivalent Version of Algorithm 1

Recall the One-epoch-SNVRG algorithm in Algorithm 1. Here we present an equivalent version of Algorithm 1 using nested loops, which is displayed in Algorithm 4 and is more aligned with the illustration in Figure 2. Note that the notation used in Algorithm 4 is slightly different from that in Algorithm 1 to avoid confusion.

Table 4: Parameter settings of all algorithms on SVHN dataset.

| Algorithm | With Learning Rate Decay | | | Without Learning Rate Decay | | |
|---|---|---|---|---|---|---|
| | Initial learning rate $\eta$ | Batch size $B$ | Batch size ratio $b$ | learning rate $\eta$ | Batch size $B$ | Batch size ratio $b$ |
| SGD | 0.1 | 2048 | N/A | 0.01 | 1024 | N/A |
| SGD-momentum | 0.01 | 2048 | N/A | 0.01 | 2048 | N/A |
| ADAM | 0.001 | 1024 | N/A | 0.001 | 512 | N/A |
| SCSG | 0.01 | 512 | 4 | 0.1 | 1024 | 4 |
| SNVRG | 0.01 | 512 | 8 | 0.01 | 512 | 4 |

---

**Algorithm 4** One-epoch SNVRG($F, \mathbf{x}_0, K, M, \{T_i\}, \{B_i\}, B$)

---

1: **Input:** Function $F$, starting point $\mathbf{x}_0$, loop number $K$, step size parameter $M$, loop parameters $T_i, i \in [K]$, batch parameters $B_i, i \in [K]$, base batch $B > 0$.
   **Output:** $[\mathbf{x}_{\text{out}}, \mathbf{x}_{\text{end}}]$
2: $T \leftarrow \prod_{l=1}^{K} T_l$
3: Uniformly generate index set $I \subset [n]$ without replacement
4: $\mathbf{g}_{[t_0]}^{(0)} \leftarrow \frac{1}{B} \sum_{i \in I} \nabla f_{i_d}(\mathbf{x}_0)$
5: $\mathbf{x}_{[0]}^{(l)} \leftarrow \mathbf{x}_0, \quad 0 \leq l \leq K,$
6: **for** $t_1 = 0, \ldots, T_1 - 1$ **do**
7:     Uniformly generate index set $I \subset [n]$ without replacement, $|I| = B_1$
8:     $\mathbf{g}_{[t_1]}^{(1)} \leftarrow \frac{1}{B_1} \sum_{i \in I} \left[ \nabla f_i(\mathbf{x}_{[t_1]}^{(1)}) - \nabla f_i(\mathbf{x}_{[0]}^{(0)}) \right]$
9:     $\ldots$
10:     **for** $t_l = 0, \ldots, T_l - 1$ **do**
11:         Uniformly generate index set $I \subset [n]$ without replacement, $|I| = B_l$
12:         $\mathbf{g}_{[t_l]}^{(l)} \leftarrow \frac{1}{B_l} \sum_{i \in I} \left[ \nabla f_i(\mathbf{x}_{[t_l]}^{(l)}) - \nabla f_i(\mathbf{x}_{[t_{l-1}]}^{(l-1)}) \right]$
13:         $\ldots$
14:         **for** $t_K = 0, \ldots, T_K - 1$ **do**
15:             Uniformly generate index set $I \subset [n]$ without replacement, $|I| = B_K$
16:             $\mathbf{g}_{[t_K]}^{(K)} \leftarrow \frac{1}{B_K} \sum_{i \in I} \left[ \nabla f_i(\mathbf{x}_{[t_K]}^{(K)}) - \nabla f_i(\mathbf{x}_{[t_{K-1}]}^{(K-1)}) \right]$
17:             Denote $t = \sum_{j=1}^{K} t_j \prod_{l=j+1}^{K} T_l$, then let $\mathbf{x}_{t+1} \leftarrow \mathbf{x}_t - 1/(10M) \cdot \sum_{l=0}^{K} \mathbf{g}_{[t_l]}^{(l)}$
18:             $\mathbf{x}_{[t_K+1]}^{(K)} \leftarrow \mathbf{x}_{t+1}$
19:         **end for**
20:         $\ldots$
21:         $\mathbf{x}_{[t_l+1]}^{(l)} \leftarrow \mathbf{x}_{[T_{l+1}]}^{(l+1)}$
22:     **end for**
23:     $\ldots$
24:     $\mathbf{x}_{[t_1+1]}^{(1)} \leftarrow \mathbf{x}_{[T_2]}^{(2)}$
25: **end for**
26: $\mathbf{x}_{\text{out}} \leftarrow$ a uniformly random choice from $\{\mathbf{x}_0, \ldots, \mathbf{x}_{T-1}\}$
27: **return** $[\mathbf{x}_{\text{out}}, \mathbf{x}_T]$

---