[Reviews · NeurIPS 2018]

Reviewer 1



The paper proposes a stochastic nested variance reduced gradient descent method for non-convex finite-sum optimization. It has been studied that variance reduction in stochastic gradient evaluations improves the complexity of stochastic gradient evaluations. A popular method is stochastic variance reduced gradient (SVRG), which uses a single reference point to evaluate the gradient. Inspired by this, authors introduce variance reduction using multiple reference points with nested scheme. More precisely, each reference point updates in every T steps and the proposed algorithm uses K points and hence one-epoch iterates T^K loops. Authors demonstrate that gradient complexity of their algorithm is better than the original SVRG and stochastically controlled stochastic gradient (SCSG) under mild conditions. They also provide an algorithm and analysis for gradient dominated case, where the global optimal can be found. Finally, their algorithms show better results for practical nonconvex problems. Although the idea of this paper seems to be natural, its result is very important in the sense of good theoretical analysis and experimental justifications. This paper is well written and easy to read. Hence, I believe that this paper should be accepted in NIPS. One confusion is that it would be better if authors provide running time plot in experiments. Some readers may not satisfy with epochs-error plots. It makes this work to be more strengthen. ***** I have read the author feedback. I appreciate authors for providing the details about my questions. *****

Reviewer 2



Overall, I appreciate this submission. It does almost all it can do in the theory part and also carries some experiments. My main concern is that while this paper does a good job under the topic of SVRG, the overall contribution may not be big. Some extra comments: 1. Do authors use batch normalization and dropout in their experiments? Hows the testing accuracy compared with the state-of-art method on MNIST, CIFAR, and SVHN datasets? Testing accuracy is more important than training and validation loss. The authors want to show their method is strong even on deep networks, however, the current form of Section 5.2 looks more like a toy example. The authors can use inception or Resnet instead. Besides, they can also compared their testing accuracy with the one obtained by Adam or adagrad. -------- I've read author's feedback, I am ok with their reply.

Reviewer 3



This paper talks about new method based on first order variance reduced optimization called Stochastic Nested Variance Reduced Gradient Descent (SNVRG). It provides better convergence rate than the current state-of-the-art. Main novelty comes from usage K+1 reference points for variance reduction. SVRG can be special case of SNVRG, but analysis, which was used in this paper, generally can't due to requirement B=n(^)2C_1^2\sigma^2/\eps^2, which usually implies K>1, thus this does not provide better rate for SVRG. On the other hand, it shows that n^(2/3) can be improved to n^(1/2) in the rate, which brings huge contribution to non-convex smooth optimization. This work also uses additional assumption on the upperbound of the stochastic gradient variance, which then provides better rate comparing to SDCA. For the experiment part. Parameter settings does not follow theory, but instead grid search is used, which due to high number of parameters to be tuned makes this algorithm hard to use in practice. On the other hand, it outperforms other algorithms substantially, but it would be hard to reproduce the result since authors did not include values for stepsizes and also grid search values. There are some minors mistakes in the proofs, for instance T = \prod_{l=1}^{k} T_l = (B/2)^(1/2) instead B^(1/2). (A.1) *** I've read the author feedback ***